# MoCaE: Mixture of Calibrated Experts Significantly Improves Object Detection

**Kemal Oksuz**                                             *kemal.oksuz@five.ai*
*Five AI Ltd., United Kingdom*

**Selim Kuzucu**                                            *selim.kuzucu2@five.ai*
*Five AI Ltd., United Kingdom*

**Tom Joy**                                                 *tom.joy@five.ai*
*Five AI Ltd., United Kingdom*

**Puneet K. Dokania**                                       *puneet.dokania@five.ai*
*Five AI Ltd., United Kingdom*

**Reviewed on OpenReview:** *https://openreview.net/forum?id=fJEsas1z8J*

## Abstract

Combining the strengths of many existing predictors to obtain a Mixture of Experts which is superior to its individual components is an effective way to improve the performance without having to develop new architectures or train a model from scratch. However, surprisingly, we find that naïvely combining off-the-shelf object detectors in a similar way to Deep Ensembles, can often lead to degraded performance. We identify that the primary cause of this issue is that the predictions of the experts do not match their performance, a term referred to as miscalibration. Consequently, the most confident detector dominates the final predictions, preventing the mixture from leveraging all the predictions from the experts appropriately. To address this, when constructing the Mixture of Experts for object detection, we propose to combine their predictions in a manner which reflects the individual performance of the experts; an objective we achieve by first calibrating the predictions before filtering and refining them. We term this approach the Mixture of Calibrated Experts (MoCaE) and demonstrate its effectiveness through extensive experiments on 5 different detection tasks, showing that it: (i) improves object detectors on COCO and instance segmentation methods on LVIS by **up to** $\sim 2.5$ **AP**; (ii) reaches **state-of-the-art** on COCO test-dev with 65.1 AP and on DOTA with 82.62 $AP_{50}$; (iii) outperforms single models consistently on recent detection tasks such as Open Vocabulary Object Detection. Code is available at: https://github.com/fiveai/MoCaE.

## 1 Introduction

Deep Ensembles (DEs) (Lakshminarayanan et al., 2017) is an effective method for obtaining improved performance by simply training multiple models before combining their predictions at inference time. Providing that compute is accessible, and inference time is not a significant issue, this approach provides a significant boost in performance at minimal cost. Another variant of this approach, the Mixture of Experts (MoE) is in practice achieved by combining different predictors (Jacobs et al., 1991; Jordan & Jacobs, 1994; Xu et al., 1994; Yuksel et al., 2012; Sukhbaatar et al., 2024). Given that these experts will typically behave differently for different data samples, one would thus expect that the model is able to leverage the benefits of one whilst ignoring the contributions of the other poorer models. Interestingly, when considering object detectors, we

---

*SK and TJ contributed to this project during their employment at Five AI Oxford team.

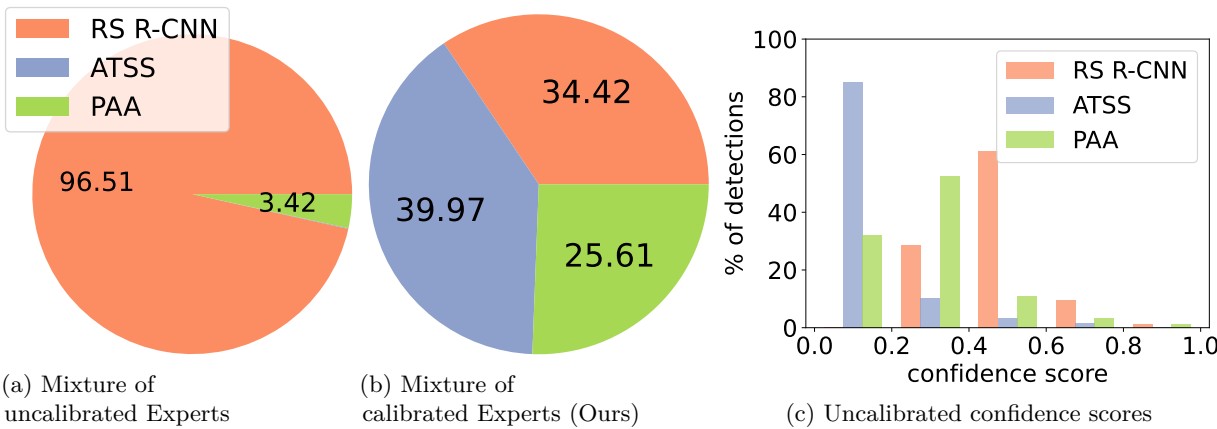

Figure 1: Piecharts showing % of detections from three similarly performing detectors in their resulting MoEs on COCO dataset. **(a)** MoE of uncalibrated detectors, **(b)** MoE of calibrated detectors, and **(c)** histogram of confidence scores.

observe that naïvely combining experts in the standard way often leads to a degradation in performance, resulting in an MoE that is completely unable to leverage the strengths of the individual experts in certain situations.

We identify that the primary reason for this is due to a failure when combining the predictions, such that the final output *does not* respect the individual performance of the experts, an issue known as miscalibration (Guo et al., 2017) implying that the predicted confidences do not match the accuracy. This inconsistency results in the most confident detector dominating the final predictions, regardless of its accuracy. To illustrate, we combine RS R-CNN, ATSS and PAA with different characteristics making them non-trivial to combine. Specifically, RS R-CNN (Oksuz et al., 2021a) is a two-stage detector optimizing a ranking-based loss function, whereas ATSS (Zhang et al., 2020b) and PAA (Kim & Lee, 2020) are both one-stage detectors both trained by focal loss (Lin et al., 2020) using different anchor-to-object assignment approaches. As can be seen in Fig. 1(a), RS R-CNN dominates the predictions due to its high level of confidence, which are shown in Fig. 1(c).

It is natural to ask why is this specific to MoE? and not present in DE? For DE, the main source of variation stems from the initialisation and other stochastic processes present in the optimisation, leading to similar histograms of predictive confidences. However, for an MoE, despite the fact that the experts perform similarly, there is a vast diversity in the mechanisms to arrive at the predictions in object detection: such as the use of an additional auxiliary localisation head (Tian et al., 2019; Zhang et al., 2020b; Jiang et al., 2018; Huang et al., 2019; Kim & Lee, 2020) or the choice of classifier, which commonly vary between a softmax (Ren et al., 2017; Carion et al., 2020; Dai et al., 2016; Bolya et al., 2019) or sigmoid classifiers for each class (Zhang et al., 2020b; Lin et al., 2020; Kim & Lee, 2020; Zhu et al., 2021). Furthermore, different backbones (Pinto et al., 2022), loss functions (Mukhoti et al., 2020) and the training length (Mukhoti et al., 2020; Oksuz et al., 2023) can drastically affect the confidence of the model.

Consequently, given the vast diversity of detectors, and the corresponding differences in their associated confidences for the predictions, it is imperative that for an effective MoE to be constructed, their confidences must match their performance; that is, they are said to be calibrated (Guo et al., 2017; Oksuz et al., 2023). To address this, we propose Mixture of Calibrated Experts (MoCaE), which calibrates the individual experts and combines the predictions using our refinement strategy, an approach we term as Refining Non-Maximum Suppression (NMS). As post-hoc calibrators are more effective compared to the existing training-time approaches (Munir et al., 2022; Pathiraja et al., 2023; Munir et al., 2023a;b) for object detection (Kuzucu et al., 2024; Oksuz et al., 2023), we employ Isotonic Regression (IR) Zadrozny & Elkan (2002) or Linear Regression (LR) as post-hoc calibrators while calibrating the experts. This makes our method an example of a practical use-case as well as a benefit of calibrating of object detectors. Furthermore, utilizing the benefits of post-hoc calibrators, **MoCaE is extremely simple to implement and leverage while combining off-the-shelf detectors.** The resulting effects of our method can be seen in Fig. 1(b) and in Fig. 2, in which

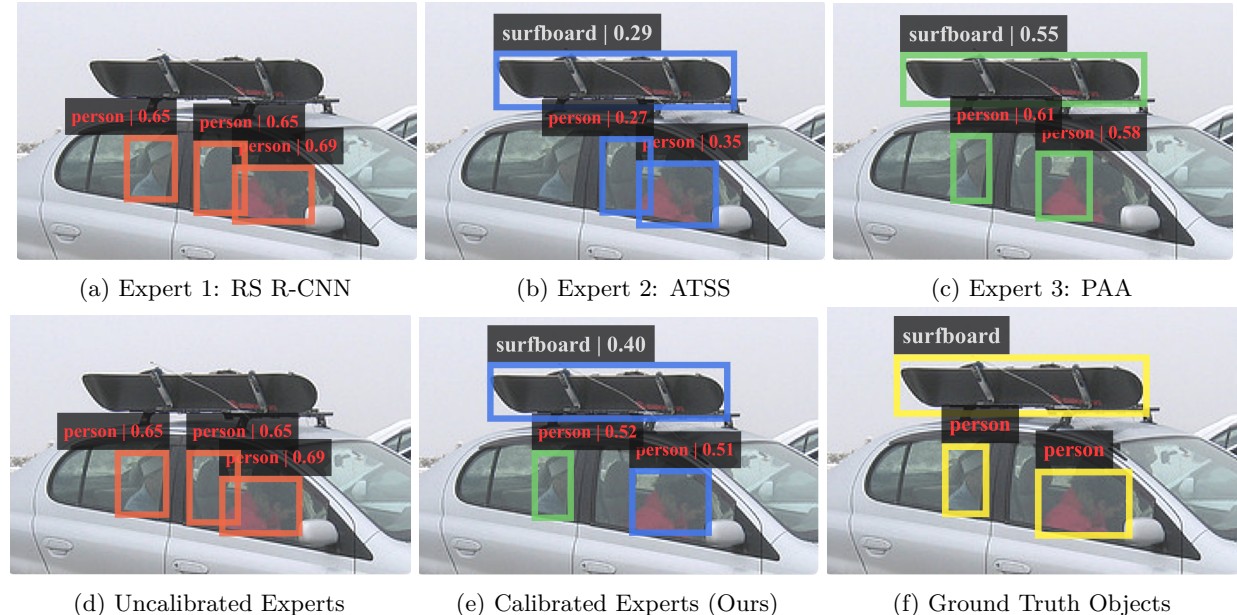

Figure 2: Detections are color-coded. red: RS R-CNN, blue: ATSS, green: PAA. **(a-c)** Outputs of the detectors on an example image. RS R-CNN misses the "surfboard", ATSS misses a "person", PAA has a notable localisation error for the "person" in front seat. **(d-f)** The detections from MoE of uncalibrated detectors; MoE of calibrated detectors; and the ground truth. (d) is dominated by the most confident RS R-CNN and misses the "surfboard". After calibration in (e), all objects are detected accurately by improving each expert.

MoCaE is able to detect all objects in the scene with good localisation quality and avoids the false-positive (FP) of the third person picked up by RS R-CNN and ATSS, but not by PAA. Overall, our contributions can be summarized as:

- We show that due to the diversity in training regimes for different detectors they consequently become miscalibrated in vastly different ways, resulting in MoEs where the most confident expert dominates.
- To address this, we propose MoCaE which first calibrates the experts and then combines their predictions through Refining NMS as our refinement mechanism while making the prediction.
- We show that MoCaE yields significant gain over the single models and DEs on different real world challenging detection tasks: such as (i) improving object detectors **by up to** $\sim 2.5$ **AP**; (ii) **reaching state-of-the-art (SOTA)** on COCO *test-dev* with 65.1 AP and on DOTA for rotated object detection with 82.62 AP; (iii) outperforming single models consistently on recent detection tasks such as Open Vocabulary Object Detection (OVOD).

## 2 Background and Notation

Given that the set of $M$ objects in an image $X$ is represented by $\{b_i, c_i\}^M$ where $b_i \in \mathbb{R}^4$ is a bounding box and $c_i \in \{1, \ldots, K\}$ its class; the goal of an object detector is to predict the bounding boxes and the class labels for the objects in $X$, $f(X) = \{\hat{c}_i, \hat{b}_i, \hat{p}_i\}^N$, where $\hat{c}_i, \hat{b}_i, \hat{p}_i$ represent the class, bounding box and confidence score of the $i$th detection respectively and $N$ is the number of predictions. In general, the detections are obtained in two steps, $f(X) = (h \circ g)(X)$ (Ren et al., 2017; Lin et al., 2020; Carion et al., 2020; Sun et al., 2018): where $g(X) = \{\hat{b}_i^{raw}, \hat{p}_i^{raw}\}^{N^{raw}}$ is a neural network predicting raw detections with bounding boxes $\hat{b}_i^{raw}$ and predicted class distribution $\hat{p}_i^{raw}$. Then, $h(\cdot)$ consists of post-processing steps to obtain the final detections from raw detections. Commonly, $h(\cdot)$ includes discarding the detections predicted as background; NMS to remove the duplicates; and keeping useful detections, normally achieved via top-$k$ survival, where typically $k = 100$ for COCO. Further discussion is provided in App. A.

Table 1: Localisation-aware Expected Calibration Error (LaECE) of different detectors on COCO *mini-test* before and after calibration using Isotonic Regression as the post-hoc calibrator. Isotonic Regression improves the calibration of different detectors.

| Calibration | RS R-CNN | ATSS | PAA |
|:---:|:---:|:---:|:---:|
| ✗ | 36.45 | 5.01 | 11.23 |
| ✓ | **3.19** | **4.54** | **1.09** |

Table 2: Recall@IoU (R@IoU where IoU is 0.50 or 0.75) and Average Recall (AR) using 100 detections for the setting in Tab. 1.

| Method | R@0.50 | R@0.75 | AR |
|:---|:---:|:---:|:---:|
| RS R-CNN (1) | 83.2 | 62.7 | 58.3 |
| ATSS (2) | 83.1 | 65.9 | 60.8 |
| PAA (3) | 83.4 | 65.8 | 61.1 |
| Uncal. MoE (1+2+3) | 83.6 | 64.1 | 59.7 |
| Cal. MoE (1+2+3) | **85.1** | **67.7** | **62.3** |

## 3 Enabling Accurate MoEs via MoCaE

We seek to examine the inconsistency of calibration errors for different detectors before proceeding to propose MoCaE. Specifically, in Sec. 3.1, we highlight the many reasons why detectors differ significantly in their confidence, and the consequences of this when constructing an MoE. To address this, Sec. 3.2 proposes MoCaE, which calibrates the individual detectors and refine their predictions.

### 3.1 Why do different detectors produce vastly different confidences?

Among different factors causing this difference, one major factor is related to the parameterisation of the predictive function. For example, some recent detectors employ an additional auxiliary head to predict localisation confidence (Tian et al., 2019; Zhang et al., 2020b; Jiang et al., 2018; Huang et al., 2019; Kim & Lee, 2020). Consequently, the auxiliary head choice such as centerness (Tian et al., 2019; Zhang et al., 2020b) or IoU (Jiang et al., 2018; Kim & Lee, 2020) as well as the aggregation function such as multiplication (Tian et al., 2019; Zhang et al., 2020b) or geometric mean (Kim & Lee, 2020) provides significant variation in the confidence scores. Architectural difference can also manifest itself in the type of the detector, which can be fully convolutional one-stage (Zhang et al., 2020b; Tian et al., 2019), two-stage (Ren et al., 2017; Cai & Vasconcelos, 2018), bottom-up (Law & Deng, 2018; Duan et al., 2019) as well as transformer-based (Carion et al., 2020; Zhu et al., 2021). Another major factor causing the confidence incompatibility across the detectors is the used classifier, which is commonly vary between a softmax (Ren et al., 2017; Carion et al., 2020; Dai et al., 2016; Bolya et al., 2019) or sigmoid classifiers for each class (Zhang et al., 2020b; Lin et al., 2020; Kim & Lee, 2020; Zhu et al., 2021). Besides, different backbones (Pinto et al., 2022), training objectives (Mukhoti et al., 2020; Chen et al., 2020; Oksuz et al., 2020; Wang et al., 2020; Lin et al., 2020; Lin et al., 2017b; Li et al., 2020; 2019; Kahraman et al., 2023; Oksuz et al., 2021b) and the training length (Mukhoti et al., 2020; Oksuz et al., 2023) affect the confidence of the model.

**How different are the predicted confidences?** To evaluate this, we leverage the recent work of Oksuz et al. (2023), requiring the confidence of the predictions to match the product of classification (cls.) and localisation (loc.) performance (perf.) using the following criterion:

$$\underbrace{\mathbb{P}(\hat{c}_i = c_i | \hat{p}_i)}_{Precision\ as\ cls.\ perf.} \underbrace{\mathbb{E}_{\hat{b}_i \in B_i(\hat{p}_i)}[\mathrm{IoU}(\hat{b}_i, b_{\psi(i)})]}_{Expected\ IoU\ as\ loc.\ perf.} = \hat{p}_i, \forall \hat{p}_i \in [0,1], \tag{1}$$

where $B_i(\hat{p}_i)$ is the set of true-positive (TP) boxes with the confidence score of $\hat{p}_i$, and $b_{\psi(i)}$ is the ground-truth box that $\hat{b}_i$ matches with. In practice, the confidence space is split into $J$ bins to compute the calibration error based on equation 1. Then, LaECE (Oksuz et al., 2023) measures the absolute difference between the accuracy and the associated confidence:

$$\sum_{j=1}^{J} \frac{|\hat{\mathcal{D}}_j^c|}{|\hat{\mathcal{D}}^c|} \left| \bar{p}_j^c - \mathrm{precision}^c(j) \times \bar{\mathrm{IoU}}^c(j) \right|, \tag{2}$$

where $\hat{\mathcal{D}}^c$ and $\hat{\mathcal{D}}_j^c$ denote the set of all detections and those in the $j$th-bin for class $c$ respectively. Similarly, the average confidence, precision and average Intersection-over-Union (IoU) of $\hat{\mathcal{D}}_j^c$ are denoted by $\bar{p}_j^c$, $\mathrm{precision}^c(j)$

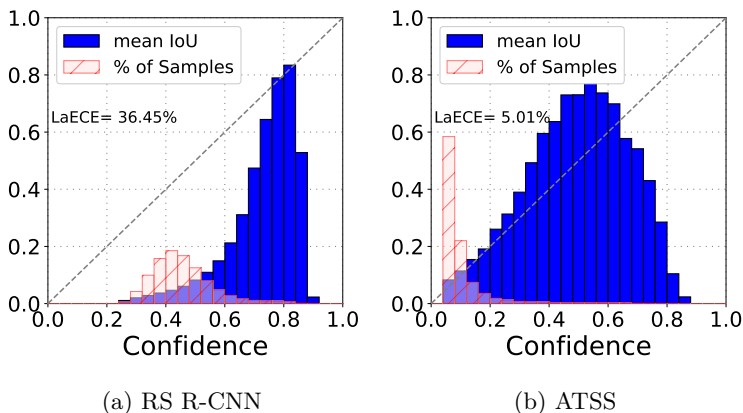

(a) RS R-CNN                           (b) ATSS

Figure 3: Reliability diagrams of **(a)** overconfident RS R-CNN and **(b)** underconfident ATSS.

and $\bar{\text{IoU}}^c(j)$. While, we employ LAECE, we slightly modify it due to the difference of our calibration criterion in Eq. 4, which will be introduced shortly. Specifically, we measure the difference between the confidence of the detection and its IoU with the best overlapping ground-truth as LAECE:

$$\text{LAECE}^c = \sum_{j=1}^{J} \frac{|\hat{\mathcal{D}}_j^c|}{|\hat{\mathcal{D}}^c|} \left| \bar{p}_j^c - \bar{\text{IoU}}^c(j) \right|. \tag{3}$$

We report LAECE in Tab. 1 where we see that RS R-CNN, PAA and ATSS, three similar performing detectors in terms of AP, have vastly different LAECE, implying different confidence predictions. Moreover, in Fig. 3, we display the reliability plots for RS R-CNN and ATSS, which shows that RS R-CNN is significantly more confident than ATSS.

**An uncalibrated Mixture of Experts** For the purposes of exposition, we first show that naïvely combining these uncalibrated detectors results in a poor MoE, where the most confident detector dominates the MoE regardless of its accuracy. To show this, similar to (Casado-García & Heras, 2020), we construct a "Vanilla MoE" which aggregates the *uncalibrated* predictions from RS R-CNN (Oksuz et al., 2021a), ATSS (Zhang et al., 2020b) and PAA (Kim & Lee, 2020) using NMS. Fig. 1(a) shows that Vanilla MoE is dominated by the most confident RS R-CNN with a small contribution from less confident PAA and almost no detections from the least confident ATSS. Furthermore, we see in Fig. 2(d), that it is unable to identify the `surfboard` and produces a FP for a person. As a result, while one would expect an MoE to detect more objects than individual detectors and obtain a better recall, Tab. 2 shows the opposite; *Vanilla MoE yields a lower Average Recall (AR) compared to ATSS and PAA.* This clearly indicates that naïvely obtaining MoE will normally be biased and lead to an ineffective mixture.

We have now highlighted that a fundamental issue with constructing an MoE, is that a situation can often arise when one of the detector dominates the predictions. However, this is not necessarily a deficiency, as one would expect an accurate detector to dominate the predictions when combined with inaccurate ones. Conceptually, we want the MoE to combine predictions based on their performance, which can be inferred through the confidence estimates provided at test time. However, as shown above, it is imperative that these predictions are calibrated. Therefore, to appropriately construct the MoE, we calibrate the experts individually, before filtering the predictions in our refinement strategy. As we show in Sec. 4, this enables reliable contributions from each detector and an effective MoE.

## 3.2 Constructing an Effective Mixture of Experts

Here we highlight the two main components for constructing an effective MoE, obtaining similarly performing calibrated experts; and aggregating their detections in the best way possible. Overall pipeline of MOCAE is presented in Fig. 4.

### 3.2.1 Calibrating Individual Experts

Having identified the issue with the Vanilla MoE, the question naturally arises as to how we calibrate the single detectors to address its deficiency. As opposed to the classification task, object detection jointly solves both classification and regression tasks; and also involves post-processing steps that influence the accuracy of the detector. Therefore, it is not straightforward as to what objective the calibrator should have and at which stage of the pipeline it should be applied. A natural choice would be to calibrate the scores such that it helps the crucial aggregation stage (e.g., NMS). This stage does not require training and has significant impact on the accuracy of a detector.

For simplicity, let's consider the standard NMS, which groups the detections that have an IoU with the maximum-scoring detection larger than a predefined IoU threshold. Then, within that group, NMS survives the detection with the largest score and removes the remaining detections from the detection set. In such a setting, as also discussed by the recent works (Li et al., 2020; 2019; Kahraman et al., 2023; Jiang et al., 2018; Zhang et al., 2020b), the ideal confidence that should be transferred to the NMS is the IoU of the detection with the object. This will guide NMS to pick *accurately-localised detections* for the objects detected by multiple detectors. Furthermore, if an object is detected by a single less confident detector, aligning the confidence with IoU implies that the scores of the TPs are to be promoted. Thus, the TPs of a less confident detector will not be dominated by the FPs of more confident ones unlike the case in Fig. 1(a). Following this intuition, we call a detector calibrated if it yields a confidence that matches the IoU, implying

$$\mathbb{E}_{\hat{b}_i \in B_i(\hat{p}_i)}[\text{IoU}(\hat{b}_i, b_{\psi(i)})] = \hat{p}_i, \forall \hat{p}_i \in [0, 1], \tag{4}$$

where $B_i(\hat{p}_i)$ is the set of detection boxes with the confidence score of $\hat{p}_i$ and $b_{\psi(i)}$ is the ground-truth box that $\hat{b}_i$ has the highest IoU with.

From an optimization perspective, calibrating each expert to meet the criterion in Eq. 4 requires us to design an objective that maps the output confidence of each bounding box to a calibrated one. Though there can be several ways to design such an objective, we take a rather simple approach where we learn a post-hoc calibrator $\zeta_\theta : [0, 1] \rightarrow [0, 1]$ using the input-target pairs $(\{\hat{p}_i, \text{IoU}(\hat{b}_i, b_{\psi(i)})\})$ obtained on a held-out validation set. Specifically, we parameterise $\zeta_\theta(\cdot)$ as a simple LR model, containing only two learnable parameters or an IR model (Zadrozny & Elkan, 2002; Oksuz et al., 2023; Kuzucu et al., 2024). Thereby being easily applicable to any off-the-shelf detector without adding any notable overhead.

Tab. 1 and Tab. 2 show an example case using IR, where the LAECE improves and the resulting MoE has a higher recall than the uncalibrated MoE and single models. Please refer to App. B and App. D for more details. Furthermore, Theorem 1 indicates that Eq. 4 is the optimal choice in terms of Average Precision (AP) in obtaining an MoE for object detection.

**Theorem 1.** *Assume that E different experts are combined in the form of an MoE and the detection set of e-th expert for class c is denoted by $\mathcal{B}_e$. Given the union of these detections (before aggregation) is denoted by $\mathcal{B}^{raw} = \bigcup_{e=1}^{E} \mathcal{B}_e$, the MoE using perfectly calibrated experts in terms of Eq. 4 yields optimal AP for class c over any possible MoEs following Lemma 1. Accordingly, denoting the number of TPs in $\mathcal{B}^{raw}$ by $N_{TP}(\mathcal{B}^{raw})$ and the number of ground-truth objects for class c by $M > 0$, the resulting AP is $\frac{N_{TP}(\mathcal{B}^{raw})}{M}$. (Please refer to App. C for Lemma 1 and the proofs.)*

### 3.2.2 Refining NMS for Aggregating Detections

Another critical component of the MoE is aggregating the combined detections. There is a very high chance that more than one detector produces the same detection; therefore we aim to suppress these duplicate detections targeting the same object and obtain detections with high localisation quality. As aforementioned, NMS is a method that fits for this purpose and, as we observe experimentally, does provide highly competitive results. However, it is rigid in nature when removing overlapping detections, thereby not utilizing the rich information provided by multiple MoEs. To address this, we present *Refining NMS* that simply combines Soft NMS (Bodla et al., 2017) with Score Voting (Kim & Lee, 2020). Before we present the details, we note that we use Refining NMS only for object detection as extending score voting to instance segmentation and rotated object detection is not trivial, hence left as a future work.

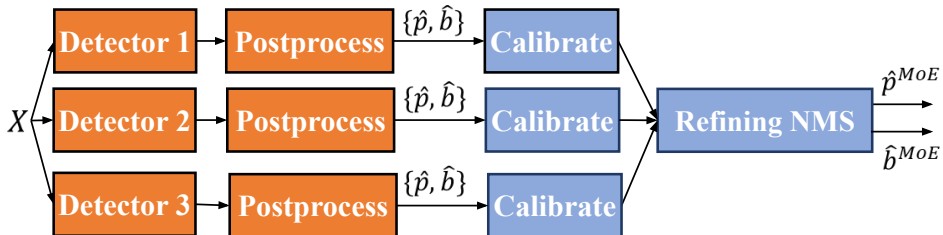

Figure 4: MoCaE pipeline. Given an image $X$, each detector follows its own pipeline including postprocessing (in orange) and outputs $\{\hat{p}, \hat{b}\}$. Without any modification to the pipeline of each detector, we calibrate the confidence scores of each detector and aggregate them via Refining NMS providing the detections of MoCaE.

Given the set of detections from all experts as $\mathcal{B} = \{(\hat{b}_i, \hat{p}_i)\}_{i=1}^{N}$ from a class $\hat{c}_i$, NMS survives the one with highest confidence (say $(\hat{b}_\alpha, \hat{p}_\alpha)$) and removes all boxes that have IoU with $\hat{b}_\alpha$ more than a pre-defined threshold $\mathrm{IoU_{NMS}}$ from $\mathcal{B}$. This process takes place until there is no remaining detection either to be survived or removed. Differently, instead of removing the detections completely, Soft NMS decreases the confidence score of a highly overlapping detection as a function of their overlap with $\{\hat{b}_\alpha, \hat{p}_\alpha\}$. In particular, we use a linear decay while decreasing the confidence of these boxes, such that

$$\hat{p}_k = \begin{cases} \hat{p}_k & \text{if } \mathrm{IoU}(\hat{b}_k, \hat{b}_\alpha) < \mathrm{IoU_{NMS}} \\ \hat{p}_k \times (1 - \mathrm{IoU}(\hat{b}_k, \hat{b}_\alpha)) & \text{else.} \end{cases} \tag{5}$$

This less rigid removal mechanism results in higher recall. Following Soft NMS, we employ Score Voting (Kim & Lee, 2020) to obtain a bounding box with better localisation for each surviving box after Soft NMS. Specifically, the refined box $\hat{b}_i$ is obtained by considering $\mathcal{B}$ as the set of detections from all experts before Soft NMS (i.e., including the ones removed by Soft NMS):

$$\hat{b}_i = \frac{\sum_{j \in \mathcal{B}} \hat{p}_j \hat{IoU}_j \hat{b}_j}{\sum_{j \in \mathcal{B}} \hat{p}_j \hat{IoU}_j}, \text{ where } \hat{IoU}_j = e^{-\frac{(1 - \mathrm{IoU}(\hat{b}_i, \hat{b}_j))^2}{\sigma_{\mathrm{SV}}}}, \tag{6}$$

where $\sigma_{\mathrm{SV}}$ is the hyper-parameter, which we set to 0.04. As a result, combining these two approaches leads to a much effective aggregator to combine the predictions from different experts.

## 4 Experiments

In this section we seek to first outline the criticality of calibrating individual object detectors when constructing an MoE (Sec. 4.1). Second, we present the effectiveness of MoCaE on many standard detection benchmarks and show that it reaches *state-of-the-art* on COCO for object detection and DOTA for rotated bounding box detection (Sec. 4.2). Then, we highlight the reliability of MoCaE under domain shift (Sec. 4.3). Finally, we outline the limitations (Sec. 4.4) of MoCaE. Our extensive experiments with 19 different detectors on 7 datasets show that MoCaE is consistently superior to the single models, Vanilla MoE and Deep Ensembles.

### 4.1 Effect of Calibration on Obtaining MoEs

To demonstrate the effect of calibration, we use the common COCO dataset (Lin et al., 2014). Similar to (Kuppers et al., 2022), we randomly split COCO val set with 5K images into two, and use 2.5K images for testing as COCO *minitest* and the remaining 2.5K images as COCO *minival* to analyse calibration. Specifically, we find it sufficient to use 500 images for calibrating the detectors. In our experiments, we mainly use COCO-style AP and also report (i) $\mathrm{AP_{50}}$, $\mathrm{AP_{75}}$ as the APs measured at IoU thresholds 0.50 and 0.75; as well as (ii) $\mathrm{AP_S}$, $\mathrm{AP_M}$ and $\mathrm{AP_L}$ to present the accuracy on small, medium and large objects. In terms of models, here we combine RS R-CNN, ATSS and PAA with ResNet-50 (He et al., 2016) with FPN (Lin et al., 2017a) backbone. These detectors have different characteristics making them non-trivial to combine. Specifically, RS R-CNN (Oksuz et al., 2021a) is a two-stage detector optimizing a ranking-based

Table 3: Effect of calibration. All MoEs except the last row use standard NMS.

| Model Type | Calibration | Combined Detectors | | | Detection Performance | | |
|---|---|---|---|---|---|---|---|
| | | RS R-CNN | ATSS | PAA | AP | $AP_{50}$ | $AP_{75}$ |
| Single Models | N/A | ✓ | | | 42.4 | 62.1 | 46.2 |
| | N/A | | ✓ | | 43.1 | 61.5 | 47.1 |
| | N/A | | | ✓ | 43.2 | 60.8 | 47.1 |
| MoEs | ✗ | ✓ | ✓ | | 42.4 | 62.1 | 46.3 |
| | ✓ | ✓ | ✓ | | **44.1** | **63.0** | **48.4** |
| | ✗ | ✓ | | ✓ | 43.4 | 62.5 | 47.1 |
| | ✓ | ✓ | | ✓ | **44.0** | **62.7** | **47.9** |
| | ✗ | | ✓ | ✓ | 43.3 | 60.9 | 47.2 |
| | ✓ | | ✓ | ✓ | **44.4** | **62.5** | **48.5** |
| | ✗ | ✓ | ✓ | ✓ | 43.4 | 62.5 | 47.1 |
| | ✓ | ✓ | ✓ | ✓ | **44.7** | **63.1** | **48.9** |
| | ✓ + Refining NMS | ✓ | ✓ | ✓ | **45.5** | **63.2** | **50.0** |

Table 5: Effect of calibration on transformer-based detectors. All MoEs except the last row use NMS.

| Model Type | Calibration | Combined Detectors | | | Detection Performance | | |
|---|---|---|---|---|---|---|---|
| | | Co-DETR | BRS-DETR | DINO | AP | $AP_{50}$ | $AP_{75}$ |
| Single Models | N/A | ✓ | | | 49.8 | 67.6 | 54.5 |
| | N/A | | ✓ | | 50.3 | 67.4 | 54.6 |
| | N/A | | | ✓ | 50.4 | 67.9 | 54.6 |
| MoEs | ✗ | ✓ | ✓ | | 49.8 | 68.4 | 54.1 |
| | ✓ | ✓ | ✓ | | **51.0** | **69.3** | **55.1** |
| | ✗ | ✓ | | ✓ | **51.0** | **69.4** | **55.2** |
| | ✓ | ✓ | | ✓ | 50.9 | **69.4** | 55.0 |
| | ✗ | | ✓ | ✓ | 50.1 | 68.3 | 53.8 |
| | ✓ | | ✓ | ✓ | **51.5** | **69.6** | **55.3** |
| | ✗ | ✓ | ✓ | ✓ | 51.0 | 69.5 | 55.2 |
| | ✓ | ✓ | ✓ | ✓ | **51.5** | **70.0** | **55.6** |
| | ✓ + Refining NMS | ✓ | ✓ | ✓ | **51.9** | **70.0** | **56.5** |

loss function, whereas ATSS (Zhang et al., 2020b) and PAA (Kim & Lee, 2020) are both one-stage detectors trained by focal loss (Lin et al., 2020). Also, different from ATSS with a centerness head, PAA employs an IoU prediction head and they differ in obtaining the confidence score.

**Calibration is crucial for accurate MoEs** In order to highlight the effect of calibration in different settings, we construct three MoEs from pair-wise combinations of RS R-CNN, ATSS and PAA as well as one MoE that combines all three. In order to focus only on calibration, here we use the standard NMS with an IoU threshold of 0.65 as in Wang et al. (2022a). Tab. 3 presents the results of MoEs with uncalibrated and calibrated detectors. The striking observation is that *without calibration, the MoEs perform similar to the single models and calibration enables accurate MoEs for all four settings.* Specifically, using two calibrated MoEs yields ∼ 1AP gain, and using three improves AP by 1.5 compared to single models; showing the effectiveness of calibration.

Table 4: LaECE of transformer-based detectors on COCO *mini-test* before and after calibration using Isotonic Regression as the post-hoc calibrator.

| Calibration | Co-DETR | BRS-DETR | DINO |
|---|---|---|---|
| ✗ | 7.63 | 13.12 | 5.06 |
| ✓ | **6.52** | **6.38** | **4.33** |

Table 6: Object detection performance on COCO *minitest*. MoEs obtained by our MoCaE outperforms DEs significantly even with less detectors. Our gains in green are obtained compared to the best single model for each performance measure, represented as underlined.

| Model Type | Detector | AP | $AP_{50}$ | $AP_{75}$ | $AP_S$ | $AP_M$ | $AP_L$ |
|---|---|---|---|---|---|---|---|
| Single Models | RS R-CNN | 42.4 | 62.1 | 46.2 | 26.8 | 46.3 | 56.9 |
| | ATSS | 43.1 | 61.5 | 47.1 | 27.8 | 47.5 | 54.2 |
| | PAA | 43.2 | 60.8 | 47.1 | 27.0 | 47.0 | 57.6 |
| Deep Ensembles | RS R-CNN $\times$ 5 | 43.4 | 63.0 | 47.7 | 28.0 | 47.5 | 57.0 |
| | ATSS $\times$ 5 | 44.1 | 62.3 | 48.4 | 29.4 | 49.0 | 56.3 |
| | PAA $\times$ 5 | 44.4 | 62.0 | 48.4 | 28.9 | 49.0 | 59.2 |
| Mixtures of Experts | Vanilla MoE (RS R-CNN, ATSS, PAA) | 43.4 | 62.5 | 47.1 | 27.3 | 47.3 | 58.0 |
| | MoCaE (ATSS and PAA) - Ours | 44.8 | 62.4 | 49.2 | 29.4 | 49.1 | 57.6 |
| | MoCaE (RS R-CNN, ATSS, PAA) - Ours | **45.5** | **63.2** | **50.0** | **29.7** | **49.7** | **59.3** |
| | | +2.3 | +1.1 | +2.9 | +1.9 | +2.2 | +1.7 |

To further show the effect of calibration, we also combine three different transformer-based detectors:

- Co-DETR (Zong et al., 2023) utilizes conventional detectors during training Deformable DETR(Zhu et al., 2021),
- BRS-DETR (Yavuz et al., 2024) trains Co-DETR with a ranking-based loss, and
- DINO (Zhang et al., 2023) is essentially an improved version of Deformable DETR.

We obtain an IR calibrator for each detector and improve their calibration as shown in Tab. 4. Then, we combine them all and in pairs in Tab. 5, which demonstrates that calibrated MoEs perform notably better in almost all combinations. For example, the gap between calibrated and uncalibrated MoEs is 1.4 AP once BRS-DETR and DINO are combined. We note that, in the single exception of combining Co-DETR and DINO, the calibrated MoE performs similar to the uncalibrated one. This is expected as the uncalibrated confidence distribution of these two detectors are very similar (as shown in Fig. A.11(a,c)), in which case calibration does not make a difference. Finally, both Tab. 3 and Tab. 5 show that Refining NMS has a positive effect on the resulting mixture. Resulting MoCaE (i.e., using calibrated and Refining NMS) outperforms the best uncalibrated MoE (i) by $\sim 2$ AP in Tab. 3 and (ii) by $\sim 1$ AP in Tab. 5.

**Even with fewer models, MoCaE is superior to DEs** Next we compare MoCaE, Vanilla MoE and DEs. We calibrate the components of DEs while combining them, though we do not observe a significant effect from the calibration in their performance (see App. D for details); which is an expected outcome. Tab. 6 shows that the DEs perform consistently better than the single models; validating them as strong baselines. The main observation in Tab. 6 is that *combining different types of few detectors into an MoE performs significantly better than DEs.* Specifically, MoCaE with only two detectors, ATSS and PAA, outperforms all DEs, each with five components. Also, combining three detectors by MoCaE performs 1.1 AP better than its closest counterpart DE. This is because the same type of detectors make similar errors, which yields less gain once they are combined together. However, different types of detectors complement each other thanks to their diversity. Finally, MoCaE outperforms Vanilla MoE by $\sim 2$ AP in this setting. Our final model obtains 45.5 AP and outperforms the best single model in all AP variants significantly.

## 4.2 Benchmarking MoCaE on Various Tasks

In this section, we demonstrate that combining off-the-shelf detectors via MoCaE improves single detectors on various detection tasks up to 2.5 AP, which is a significant performance improvement. Our MoCaE reaches SOTA results on COCO dataset among public models and on DOTA dataset for rotated object detection. Specifically, we evaluate on four different tasks: object detection (COCO (Lin et al., 2014)), rotated object detection (DOTA (Xia et al., 2018)), open vocabulary object detection (COCO and ODinW35 (Li et al., 2022a)) and instance segmentation (LVIS (Gupta et al., 2019)). For these tasks, we use a total of 15 different detectors include one-stage and two-stage, convolutional, transformer-based ones and foundation models.

Table 7: Detection performance on COCO *test-dev* using strong detectors. Green: Gain against best single model (underlined).

| Method | AP | AP$_{50}$ | AP$_{75}$ | AP$_S$ | AP$_M$ | AP$_L$ |
|---|---|---|---|---|---|---|
| YOLOv7 (Wang et al., 2022a) | 55.5 | 73.0 | 60.6 | 37.9 | 58.8 | 67.7 |
| QueryInst (Fang et al., 2021) | 55.7 | 75.7 | 61.4 | 36.2 | 58.4 | 70.9 |
| DyHead (Dai et al., 2021) | 56.6 | 75.5 | 61.8 | 39.4 | 59.8 | 68.7 |
| Vanilla MoE | 57.6 | 76.6 | 63.2 | 40.0 | 60.9 | 70.8 |
| | +1.0 | +0.9 | +1.4 | +0.6 | +1.1 | −0.1 |
| MoCaE | **59.0** | **77.2** | **64.7** | **41.1** | **62.6** | **72.4** |
| (Ours) | +2.4 | +1.5 | +2.9 | +1.7 | +2.8 | +1.5 |

Table 8: Performance on COCO *test-dev* using state-of-the-art detectors. MoCaE improves the most accurate publicly available model by 0.7 AP and reaches SOTA.

| Method | AP | AP$_{50}$ | AP$_{75}$ | AP$_S$ | AP$_M$ | AP$_L$ |
|---|---|---|---|---|---|---|
| EVA (Fang et al., 2023) | 64.4 | 82.3 | 70.9 | 48.2 | 67.6 | 77.5 |
| Co-DETR (Zong et al., 2023) | 64.3 | 81.4 | 71.0 | 48.1 | 67.1 | 77.5 |
| Vanilla MoE | 64.6 | **82.7** | 71.0 | 48.5 | 67.6 | 77.5 |
| | +0.2 | +0.4 | 0.0 | +0.3 | 0.0 | 0.0 |
| MoCaE | **65.1** | **82.7** | **71.9** | **49.2** | **68.1** | **78.1** |
| (Ours) | +0.7 | +0.4 | +0.9 | +1.0 | +0.5 | +0.6 |

Table 9: Rotated object detection performance on DOTA test set. AP$_{50}$ is used following DOTA. See App. D for all classes.

| Method | AP$_{50}$ | 5 Classes with Lowest Performance | | | | |
|---|---|---|---|---|---|---|
| | | Bridge | Soccer | Roundab. | Harbor | Helico. |
| RTMDet (Lyu et al., 2022) | 81.32 | 58.50 | 72.12 | 70.85 | 81.16 | 77.24 |
| LSKN (Li et al., 2023) | 81.85 | 61.47 | 71.67 | 71.35 | 79.19 | 80.85 |
| Vanilla MoE | 80.60 | **61.77** | 70.98 | 65.92 | 84.28 | 77.57 |
| | −1.25 | +0.30 | −1.14 | −5.43 | +3.12 | −3.28 |
| MoCaE | **82.62** | 61.38 | **75.50** | **74.12** | **84.49** | **81.93** |
| (Ours) | +0.77 | −0.09 | +3.38 | +2.77 | +3.33 | +1.08 |

Table 10: Comparison on open vocabulary object detection. Green: Gain against best single model (underlined).

| Method | COCO | | | ODinW-35 | |
|---|---|---|---|---|---|
| | AP | AP$_{50}$ | AP$_{75}$ | AP$_{avg}$ | AP$_{median}$ |
| Grounding DINO-T (Liu et al., 2024) | 48.5 | 64.5 | 52.9 | 22.7 | 13.8 |
| MQ-GLIP-T (Xu et al., 2023) | 46.3 | 62.7 | 50.6 | 21.4 | 8.2 |
| Vanilla MoE | 48.1 | 64.8 | 52.5 | 22.4 | 11.8 |
| | −0.4 | +0.3 | −0.4 | −0.3 | −2.0 |
| MoCaE | **49.5** | **66.4** | **54.3** | **23.2** | **14.9** |
| (Ours) | +1.0 | +2.1 | +1.4 | +0.5 | +1.1 |

**Object Detection on COCO** Here, we evaluate on COCO *test-dev* by submitting our result to the evaluation server. In the first setting, we combine the following well-known and effective detectors:

- YOLOv7 (Wang et al., 2022a) with a large convolutional backbone following its original setting,
- QueryInst (Fang et al., 2021) as a transformer-based detector with a Swin-L (Liu et al., 2021) backbone,
- ATSS with transformer-based dynamic head (Dai et al., 2021) and again Swin-L backbone.

These detectors differ from each other in terms of the pretraining data, backbone or architecture as summarized in App. D. Tab. 7 shows that our MoCaE reaches 59.0 AP with a gain of 2.4 AP on this challenging setting as well. As our gain here is similar to that of Tab. 6, we can easily say that the gain of our MoCaE has not saturated in this stronger setting, which is commonly the opposite in the literature.

Finally, we also evaluate MoCaE on the two most recent SOTA publicly available detectors[*]:

- EVA (Fang et al., 2023), a foundation model for vision using Cascade Mask R-CNN (Cai & Vasconcelos, 2018) for detection,
- Co-DETR (Zong et al., 2023), a transformer-based detector.

*Tab. 8 shows that MoCaE reaches SOTA with* 65.1 *AP on COCO test-dev and outperforms all existing public detectors by* 0.7 *AP.* This further shows the effectiveness of MoCaE.

**Rotated Object Detection on DOTA** We now investigate MoCaE for rotated object detection on DOTA v1.0 dataset (Xia et al., 2018) with 15 classes. DOTA is also a challenging dataset comprising of aerial images that are very dense in terms of objects. Specifically, DOTA dataset has 67.1 objects on average per image. We use all 458 images in the validation set to calibrate the detectors and report AP$_{50}$ on the test set by submitting our results to the evaluation server. We combine LSKN (Li et al., 2023) and RTMDet (Lyu et al., 2022) as two recent SOTA detectors. Following the literature, we use NMS with an IoU threshold of 0.35 as Soft NMS and Score Voting are not straightforward to use in this task. Tab. 9 suggests that *we establish a new SOTA* with 82.62 AP$_{50}$ on DOTA; improving the previous SOTA by 0.77. Having examined the classes,

---

[*]Published at CVPR 2023 and ICCV 2023

Table 11: Instance segmentation results on LVIS val set. $AP_{box}$ denotes detection AP.

| Method | AP | $AP_{50}$ | $AP_{75}$ | $AP_r$ | $AP_c$ | $AP_f$ | $AP_{box}$ |
|---|---|---|---|---|---|---|---|
| Mask R-CNN | 25.4 | 39.2 | 27.3 | 15.7 | 24.7 | 30.4 | 26.6 |
| RS Loss | 25.1 | 38.2 | 26.8 | 16.5 | 24.3 | 29.9 | 25.8 |
| Seesaw Loss | 25.4 | 39.5 | 26.9 | 15.8 | 24.7 | 30.4 | 25.6 |
| Vanilla MoE | 25.2 | 38.3 | 26.8 | 16.5 | 24.3 | 29.9 | 25.9 |
| | −0.2 | −1.2 | −0.5 | 0.0 | −0.4 | −0.5 | −0.7 |
| MoCaE (Ours) | **27.7** | **42.8** | **29.4** | **18.2** | **27.3** | **32.4** | **29.1** |
| | +2.3 | +3.3 | +2.1 | +1.7 | +2.4 | +2.0 | +2.5 |

Table 12: Instance segmentation results on COCO *minitest* set combining transformer-based Mask2Former and VitDet with Mask R-CNN.

| Method | AP | $AP_{50}$ | $AP_{75}$ | $AP_s$ | $AP_m$ | $AP_l$ |
|---|---|---|---|---|---|---|
| Mask2Former | 46.6 | 69.5 | 50.4 | 26.7 | 50.1 | **70.5** |
| ViTDet | 46.2 | 69.5 | 50.5 | 28.1 | 49.9 | 65.5 |
| Vanilla MoE | 47.3 | 71.4 | 51.6 | 28.9 | 50.8 | 68.7 |
| | +0.7 | +1.9 | +0.8 | +2.2 | +0.7 | −1.8 |
| MoCaE (Ours) | **47.7** | **71.7** | **52.1** | **29.2** | **51.5** | 68.7 |
| | +1.1 | +2.2 | +1.7 | +2.5 | +1.4 | −1.8 |

we note that our improvement originates mostly from the classes with relatively lower performance; with the exception of the class 'bridge' which performs marginally worse. For example, on 'soccer-field', 'roundabout', 'harbor' classes where the single detectors have between $70 − 80$ $AP_{50}$, the improvement is around 3 $AP_{50}$. These gains enable us to demonstrate the ability of MoCaE to set a new SOTA in rotated object detection.

**Open Vocabulary Object Detection (OVOD)** We now investigate the effect of MoCaE on OVOD task (Li et al., 2022b;a). OVOD task is a recently proposed challenging task in which the aim is to detect the objects pertaining to the classes in a given text prompt. This requires the models to be able to interpret the given text prompt as well as the image and yield the detection results. Furthermore, the text prompt can contain phrases that are not necessarily in the training data. For this challenging task, we combine two strong and recent models:

- Grounding DINO (Liu et al., 2024), a transformer-based detector, and
- MQ-Det (Xu et al., 2023), an anchor-based OVOD relying on GLIP (Li et al., 2022b).

We obtain the calibrators on a subset of the Objects365 dataset (Shao et al., 2019), which is included in the pretraining data for both of these models. Therefore, the calibrators are also limited to the pretraining data only. In our evaluation, we evaluate the models on COCO and ODinW-35(Li et al., 2022a) datasets following the common convention (Li et al., 2022b; Liu et al., 2024; Xu et al., 2023). Note that ODinW-35 is a challenging dataset with 35 different subdatasets, some of which are substantially different from the pretraining dataset. To illustrate, ODinW-35 includes subdatasets specifically for 'potholes' on the road and *infrared* images of 'dogs' and 'people'. Tab. 10 shows that MoCaE improves the single models on both of these datasets on all performance measures notably. As an example, the median AP of the best single model on ODinW-35 increases from 13.8 to 14.9, which suggests an 8% relative gain.

**Instance Segmentation** Given that MoCaE is beneficial for object detection, one would expect it to improve performance on the instance segmentation task. To verify this, we first use LVIS (Gupta et al., 2019) as a long-tailed dataset for instance segmentation with more than 1K classes. Following its standard evaluation, we also report the AP on rare ($AP_r$), common ($AP_c$) and frequent ($AP_f$) classes. Similar to COCO, we reserve 500 images from val set to calibrate the detectors, and test our models on the remaining 19.5K images of the val set. We combine three diverse off-the-shelf Mask R-CNN variants in a MoE:

- The vanilla Mask R-CNN (He et al., 2017) with ResNeXt-101 (Xie et al., 2016) backbone, softmax classifier and using Repeat Factor Sampling (RFS) used for the long-tailed nature of LVIS,
- Mask R-CNN with ResNet-50, sigmoid classifier, trained with RS Loss (Oksuz et al., 2021a) and RFS,
- Mask R-CNN with ResNet-50, softmax classifier, trained with Seesaw Loss(Wang et al., 2020) but no RFS.

Tab. 11 shows that *while the Vanilla MoE performs worse than the best single model, MoCaE boosts the segmentation AP by* 2.3, an improvement of $∼ 10\%$ over the best single model. Also, the detection AP ($AP_{box}$) improves by 2.5 aligned with our previous findings.

Table 13: Comparison on domain-shifted datasets using COCO *mini-test* with ImageNet-C style corruptions and Objects45K, a natural shift from COCO. Green: Gain against best single model (underlined).

| Model Type | Detector | Corrupted COCO *mini-test* | | | | | | Objects45K | | |
| | | Severity of the Corruption | | | | | mean | | | |
| | | 1 | 2 | 3 | 4 | 5 | AP | AP | $AP_{50}$ | $AP_{75}$ |
| Single Models | RS R-CNN | 33.2 | 27.7 | 21.9 | 16.2 | 11.8 | 22.2 | 28.6 | 41.2 | 31.2 |
| | ATSS | 33.8 | 28.2 | 22.2 | 16.3 | 11.8 | 22.5 | 28.7 | 39.8 | 31.2 |
| | PAA | 34.4 | 29.2 | 23.1 | 16.9 | 12.1 | 23.1 | 28.7 | 39.5 | 31.0 |
| DEs | RS R-CNN × 5 | 34.6 | 29.3 | 23.5 | 17.5 | 12.8 | 23.5 | 29.8 | **42.6** | 32.7 |
| | ATSS × 5 | 35.0 | 29.5 | 23.4 | 17.4 | 12.7 | 23.6 | 29.3 | 40.4 | 31.8 |
| | PAA × 5 | 35.7 | 30.4 | 24.4 | **18.2** | **13.2** | 24.4 | 29.4 | 40.1 | 31.8 |
| MoEs | Vanilla MoE | 34.5 | 29.1 | 23.1 | 17.2 | 12.5 | 23.3 | 29.3 | 41.4 | 31.8 |
| | MoCaE (Ours) | **36.3** +1.9 | **30.8** +1.6 | **24.6** +1.5 | **18.2** +1.3 | **13.2** +1.1 | **24.6** +1.5 | **30.6** +1.9 | 41.6 +0.4 | **33.3** +2.1 |

Secondly, we combine two recent highly accurate approaches on COCO dataset to further support our claims:

- Mask2Former (Cheng et al., 2022) with Swin-S Liu et al. (2021) backbone and
- VitDet(Li et al., 2022c) that uses a Mask R-CNN prediction head with a transformer-based Vit-B backbone (Dosovitskiy et al., 2021).

Tab. 12 shows that MoCaE improves the best detector by 1.1 mask AP, outperforms Vanilla MoE as well, thereby showing the effectiveness of MoCaE for combining instance segmentation models.

## 4.3 How Reliable is MoCaE?

As we are ensembling detectors in the form of MoE, it is naturally to evaluate how reliable MoCaE is. To evaluate this, we investigate how MoCaE performs under domain shift (Michaelis et al., 2019; Hendrycks & Dietterich, 2019), and also test MoCaE on the recently proposed Self-aware Object Detection (SAOD) task (Oksuz et al., 2023). Here, we use our setting in Sec. 4.1, in which we combine RS R-CNN, ATSS and PAA. That is, we use the calibrators trained on clean COCO and do not train a new calibrator.

**Domain Shift (Synthetic and Natural)** Following the convention (Michaelis et al., 2019; Oksuz et al., 2023; Munir et al., 2022), we apply 15 ImageNet-C style corruptions (Hendrycks & Dietterich, 2019) under 5 different severities for synthetic domain shift on COCO. Similar to the clean data, we observe in Tab. 13 that combining only three models (RS R-CNN, ATSS and PAA) outperforms DEs with five components thanks to the diversity of the detectors. Here we see, that the performance over the best single model improves by 1.5 AP. Next, we evaluate the performance of MoCaE on Objects45K (Oksuz et al., 2023). Please note that this dataset has the same set of classes as in COCO, but collected and annotated separately, thereby implying a natural domain shift and a similar setting was used in (Harakeh & Waslander, 2021). Tab. 13 shows that the DEs do not provide notable gains in this setting. For example, while PAA × 5 only improves single PAA only by 0.7 AP, MoCaE improves the best single model by ∼ 2 AP, outperforming all DEs and Vanilla MoE.

**Self-aware Object Detection (SAOD)** Finally, we evaluate MoCaE on the recently proposed SAOD task (Oksuz et al., 2023), which requires the object detectors to be self-aware. This task requires detectors to provide reliable uncertainty estimates along with accurate and calibrated detections in a holistic manner; which is evaluated using the Detection Awareness Quality (DAQ). To evaluate the models on this task, we convert RS R-CNN, ATSS, PAA, Vanilla MoE and MoCaE to a self-aware detector following (Oksuz et al., 2023). MoCaE improves the DAQ of the best single model from 40.9 to 42.9 and outperforms Vanilla MoE by ∼ 0.5 DAQ. Overall, this suggest that MoCaE is more reliable than the single detectors. Details are provided in App. D due to space limitation.

### 4.4 Ablation Analysis, Discussion and Limitations

**Ablation Analysis** We provide further ablation of MoCaE in Tab. 14 in which the calibration appears to be the major factor of the performance gain. Soft NMS yields a small gain of 0.1 AP and Score Voting, combining boxes from different detectors to extract a new bounding box, improves AP notably from 44.8 to 45.5. Please refer to App.D for further details.

**Using Recent MoE Approaches to Combine Off-the-shelf Detectors** Recently-proposed MoE methods have been very effective to improve the performance of the transformer-based models (Jawahar et al., 2023; Dai et al., 2024; Lin et al., 2024; Jiang et al., 2024; Shen et al., 2023; Pióro et al., 2024; Zhou et al., 2022; Fedus et al., 2022; Ruiz et al., 2021). Conventionally, these methods replace the fully connected layer in the transformer by a softmax layer to route the input followed by multiple experts of fully connected layers. They are commonly applied to the classification-based problems and supervised during training in an end-to-end manner. Considering their success, we now aim to see whether such methods can be useful in combining multiple off-the-shelf detectors leveraging a small val. set, as we did. To do so, we design a Transformer layer that routes the detections from different experts. Specifically, given detections from all experts, this layer consists of a self-attention block in which all detections attend to each other followed by a sigmoid for each detection to predict its confidence. Similar to our approach, the predicted confidence is trained using COCO *mini-val* set to match the IoU of the detection with the ground truth and to select the set of detections yielding the highest AP. Though we perform proper hyper-parameter search and experimented with different loss functions, this model does not generalize well enough to the test set. Essentially, the AP on the data that it is trained is very high, but the test AP is significantly low. This suggests that non-trivial changes are required in order to make these MoEs effective for combining off-the-shelf object detectors.

**Limitations** In cases where the performance gap is large between the experts, we observe that both MoCaE and Vanilla MoE tend to perform worse than the best single model. For example, combining RS R-CNN, ATSS, PAA (APs $\approx$ 40) with EVA and Co-DETR (APs $\approx$ 65) does not result in a stronger MoE than EVA or Co-DETR (refer to App. D for the details). However, once we use an Oracle MoE obtained by directly assigning the calibration targets as the confidence, MoCaE reaches 86.7AP achieving more than 20 AP improvement compared to the best single model; an observation which aligns with our theoretical insights. This suggests that calibration is critical for effective MoEs and more effort is required from the community in calibrating object detectors.

Table 14: Ablation analysis of MoCaE. We use the MoE combining RS R-CNN, ATSS and PAA on COCO *mini-test*.

| Calibration | Refining NMS | | AP |
| | Soft NMS | Score Voting | |
|---|---|---|---|
| ✗ | ✗ | ✗ | 43.3 |
| ✓ | ✗ | ✗ | 44.7 |
| ✓ | ✓ | ✗ | 44.8 |
| ✗ | ✓ | ✗ | 43.4 |
| ✗ | ✓ | ✓ | 44.4 |
| ✓ | ✓ | ✓ | **45.5** |

## 5 Conclusions

A direct result of the vastly different training regimes employed in training object detectors is that their predictions are miscalibrated such that some are more confident than the others. This lack of consistency makes constructing an MoE in a naïve approach futile, as the most confident detector dominates the predictions, even though its performance may not warrant this weighting. Consequently, to address this we introduced MoCaE as a simple, principled and effective approach, which first calibrates the individual detectors before combining them appropriately through our refinement strategy. Specifically, in the calibration stage we aligned the confidence with the IoU of the detection with the object that it overlaps the most with. We showed that this is an effective calibration target, resulting in accurate MoEs with consistent gains across different detection tasks, reaching SOTA on many challenging detection benchmarks such as COCO and DOTA. Whilst our choice of calibration function performed well on the settings demonstrated here, we further observe that increased gains can be achieved if the community develops more sophisticated calibration methods, an objective we leave to future work.

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

# APPENDICES

## A  Related Work

**Mixture of Experts (MoEs)**  The main aim of combining multiple experts in the form of an MoE is to leverage the expertise of each expert, which ideally specialises in different subpopulations of the input data (Jacobs et al., 1991; Jordan & Jacobs, 1994; Xu et al., 1994; Yuksel et al., 2012). In the literature, this aim is achieved by a variety of techniques. Some of the methods aggregates the predictions of individual experts (Casado-García & Heras, 2020). Broadly speaking, from this perspective, DEs (Lakshminarayanan et al., 2017), in which multiple models are trained and then aggregated, can also be considered as an example in this group. Another set of methods (Zhou et al., 2020; Lee et al., 2020; Wang et al., 2022b; Zhang et al., 2022; Cai et al., 2021; Ruiz et al., 2021), as the majority of the techniques proposed in the deep learning era, allows the experts to share certain network components among experts, which are then combined (or selected) by a gating (a.k.a. routing) mechanism for the sake of efficiency. Essentially, this gating mechanism decides on which expert to rely on conditioned on a particular subspace of the input space.One particular and intuitive use-case of this group of methods is long-tailed classification (Zhou et al., 2020; Wang et al., 2022b; Cai et al., 2021; Zhang et al., 2022), in which different experts specialise on different classes with various cardinalities, i.e. the classes with few or many training examples. The shared features are then routed by a gating mechanism to leverage experts' specialisation in an efficient manner. Another effective application of MoEs is within the domain of variational autoencoders (Wu & Goodman, 2018; Shi et al., 2019; Joy et al., 2022), where different experts are generally utilised for different modalities.

**Ensemble Methods in Object Detection**  Despite their aforementioned success in the classification literature, ensembling detectors either in the form of a DE by using the same model with different initialisations or as an MoE by combining different type of models has received very little attention (Lee et al., 2020; Casado-García & Heras, 2020). Among the few existing works, Casado-García & Heras (2020) combine a set of detectors through various aggregation strategies, such as unanimous agreement between the detectors. Therefore, this method combines off-the-shelf detectors, and accordingly, we refer to as Vanilla MoE in the paper as a baseline. However, we note that it does not consider either calibrating the detector or advanced aggregation techniques unlike our work. As the second work in this domain, Multi-Expert R-CNN(Lee et al., 2020) is designed to exploit multiple experts in the R-CNN family, in which different R-CNN models correspond to different experts for a specific RoI. Consequently, it is not applied to a wide range of different detectors such as the common one-stage detectors or transformer-based detectors.

**Calibration in Object Detection**  As extensively studied for classification, calibration refers to the alignment of accuracy and confidence of a model (Guo et al., 2017; Nixon et al., 2019; Kumar et al., 2019; Wang et al., 2021; Mukhoti et al., 2020; Cheng & Vasconcelos, 2022). Specifically, a classifier is said to be *calibrated* if it yields an accuracy of $p$ on its predictions with a confidence of $p$ for all $p \in [0, 1]$. Earlier definitions for the calibration of detectors (Kuppers et al., 2020; Neumann et al., 2018) extend this definition with an objective to align the confidence of a detector with its precision,

$$\mathbb{P}(\hat{c}_i = c_i | \hat{p}_i) = \hat{p}_i, \forall \hat{p}_i \in [0, 1], \tag{A.7}$$

where $\mathbb{P}(\hat{c}_i = c_i | \hat{p}_i)$ denotes the precision as the ratio of correctly classified predictions among all detections. Extending from this definition, Oksuz et al. (2023) take into account that object detection is a joint task of classification and localisation. Thereby defining the accuracy as the product of precision and average IoU of TPs, calibration of object detectors requires the following to be true

$$\mathbb{P}(\hat{c}_i = c_i | \hat{p}_i)\mathbb{E}_{\hat{b}_i \in B_i(\hat{p}_i)}[\text{IoU}(\hat{b}_i, b_{\psi(i)})] = \hat{p}_i, \forall \hat{p}_i \in [0, 1], \tag{A.8}$$

where $B_i(\hat{p}_i)$ is the set of TPs with the confidence of $\hat{p}_i$ and $b_{\psi(i)}$ is the object that $\hat{b}_i$ matches with. Then, LAECE is obtained by discretizing the confidence score space into $J$ bins for each class. Specifically for class $c$, denoting the set of detections by $\hat{\mathcal{D}}^c$ and those in the $j$th bin by $\hat{\mathcal{D}}_j^c$ as well as the average confidence,

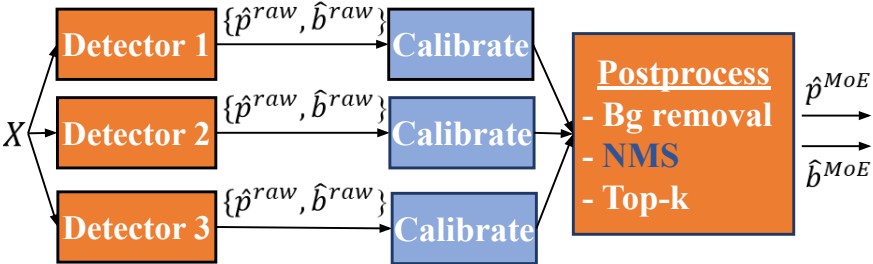

Figure A.5: Early calibration. Raw confidence scores $\hat{p}^{raw}$ are calibrated, and the standard post-processing steps handle aggregation in which NMS (in blue) removes the duplicates from multiple detectors.

precision and average IoU of $\hat{\mathcal{D}}_j^c$ by $\bar{p}_j^c$, precision$^c(j)$ and $\bar{\text{IoU}}^c(j)$ respectively, LaECE for class $c$ is defined as

$$\text{LaECE}^c = \sum_{j=1}^{J} \frac{|\hat{\mathcal{D}}_j^c|}{|\hat{\mathcal{D}}^c|} \left| \bar{p}_j^c - \text{precision}^c(j) \times \bar{\text{IoU}}^c(j) \right|. \tag{A.9}$$

Finally, the detector LaECE is the average of LaECE$^c$s over classes in the dataset, measuring the calibration error for a detector as a lower-better measure.

**Comparative Summary** Different from the existing few works on the ensemble methods of object detectors, we comprehensively investigate how to obtain MoEs in object detection using off-the-shelf detectors. While doing so, unlike existing work, (i) we identify that miscalibration of different detectors prevents them to be combined properly due to the peculiarities of the detectors; (ii) we introduce a strong aggregation technique which we refer to as Refining NMS combining Soft NMS and Score Voting from previous work; and (iii) we employ a very diverse set of detectors from a wide range of detection tasks. As for the miscalibration of the detectors, we rely on LaECE in Eq. A.8. Consequently, MoCAE follows the main aim of the MoE as the predictions from each expert are aggregated in a way that the strength of each detector is leveraged as we comprehensively demonstrate in our experiments. And besides, from the MoE perspective Refining NMS can be considered as a form of a gating (or routing) mechanism, which decides the best way to combine the detections from different experts.

# B Further Details on Calibration

Throughout our paper, we calibrate the final confidence scores. One can also consider in object detection that, the raw confidence scores (before postprocessing) can be calibrated as well. For this reason, we first present early calibration in App. B.1. Then, for the sake of completeness, we discuss why we prefer post-hoc calibration methods in App. B.2 as well as how to measure the calibration error of MoCAE based on Eq. (4) in App. B.3.

## B.1 Early Calibration of Object Detectors for Obtaining MoEs

Up to now, we discussed calibrating final confidence scores $\hat{p}_i$ similar to Kuppers et al. (2020); Oksuz et al. (2023), which we show as *late calibration* in Fig. 4. Besides, we also investigate the effectiveness of *early calibration* by calibrating the raw probabilities $\hat{p}_i^{raw}$ of the detectors as illustrated in Fig. A.5. While we find both approaches to perform similar in MoEs, we use late calibration as it is simpler owing to less number of final detections. Still, as the first to investigate early calibration, we present an additional use-case of early calibration in App. D in which we show that it reduces the sensitivity of the model to the background removal threshold in post-processing.

## B.2 Choice of the Calibration Method

There are multiple calibration methods used for object detection including training-time (Munir et al., 2022; 2023a; Pathiraja et al., 2023) and post-hoc calibration methods (Oksuz et al., 2023; Zadrozny & Elkan, 2002; Guo et al., 2017). Ideally, we would expect a calibration method to easily generalize to a wide range of detectors as well as detection tasks. However, training-time calibration methods are typically incorporated into only a small set of object detectors and not tested on a variety of relevant tasks such as the rotated bounding box detection, which makes them unfit for purposes. On the other hand, post-hoc calibration methods formalize calibration as a regression task from the predicted final confidence to target confidence, making them easily applicable to any detectors and any detection task. Throughout this work, we investigate Linear Regression (LR) and Isotonic Regression (IR) (Zadrozny & Elkan, 2002) considering the criterion in Eq. 4. *We further observe that a Class-agnostic (CA) IR calibrator for each detector obtained on 500 images is sufficient for MoCAE.* App D present extensive experimental analyses.

## B.3 Measuring the Calibration Error for MoE

We introduce our calibration criterion in Eq. 4. This criterion requires the confidence of a detection to align with its IoU with the ground truth box that the detection overlaps the most. As stated earlier, computing the calibration error based on this criterion corresponds to using an IoU threshold of 0 to validate TPs. This is equivalent to using $\text{precision}^c(j) = 1$ for class $c$ in Eq. 2, which then reduces to

$$\text{LaECE}^c = \sum_{j=1}^{J} \frac{|\hat{\mathcal{D}}_j^c|}{|\hat{\mathcal{D}}^c|} \left| \bar{p}_j^c - \bar{\text{IoU}}^c(j) \right|. \tag{A.10}$$

where $\hat{\mathcal{D}}^c$ denotes the set of detections for class $c$; $\hat{\mathcal{D}}_j^c$ is the set of detections in the $j$th bin for class $c$; $\bar{p}_j^c$ is the average confidence of the detections in $\hat{\mathcal{D}}_j^c$; and $\bar{\text{IoU}}^c(j)$ is the average IoU of the detections in $\hat{\mathcal{D}}_j^c$. Following Oksuz et al. (2023), we use $J = 25$ and average over $\text{LaECE}^c$ of classes for the detector LaECE.

In addition to LaECE, here we define Localisation-aware Average Calibration Error (LaACE) and Localisation-aware Maximum Calibration Error (LaMCE) similar to the way how Expected Calibration Error (ECE) is extended to Average Calibration and Maximum Calibration Errors. We find LaACE and LaMCE useful as they reduce the dominance of certain bins on the calibration error as in the case of LaECE. This is especially important for early calibration from which thousands of confidence scores are obtained from a single image, most of which have a confidence close to 0. Specifically, in our case, we define $\text{LaACE}^c$ for class $c$ as

$$\text{LaACE}^c = \sum_{j=1}^{J} \frac{1}{J} \left| \bar{p}_j^c - \bar{\text{IoU}}^c(j) \right|, \tag{A.11}$$

and $\text{LaMCE}^c$ for class $c$ as

$$\text{LaMCE}^c = \max_{j \in \{1,2,..,J\}} \left| \bar{p}_j^c - \bar{\text{IoU}}^c(j) \right|. \tag{A.12}$$

Following LaECE, we obtain LaACE and LaMCE for the detector by averaging over the classes.

## C  Theoretical Discussion on the Optimality of the Calibration Criterion in Eq. 4

Lemma 1 discusses the conditions under which an optimal AP can be achieved for a given set of pre-NMS detections.

**Lemma 1.** *Given a set of detection boxes for class $c$, denoted by $\mathcal{B}^{raw} = \{\hat{b}_1^{raw}, \hat{b}_2^{raw}, ..., \hat{b}_L^{raw}\}$, we first assume that the post-processing (NMS in this case) does not remove TPs and can remove duplicates in $\mathcal{B}^{raw*}$. Let us*

---

[*]A TP is a detection that has at least an IoU with a ground-truth of $\tau$ where $\tau$ is the IoU threshold to validate TPs. In the case of more than one detection satisfy this criterion, the common convention is to accept the detection with the highest score as a TP and the remaining ones are duplicates, which are counted as FPs while computing the AP.

*denote the kth ground-truth box by $b_k$ and the detection set post NMS by $\mathcal{B} = \{(\hat{b}_1, \hat{p}_1), (\hat{b}_2, \hat{p}_2)..., (\hat{b}_N, \hat{p}_N)\}$ where $(\hat{b}_i, \hat{p}_i)$ correspond to a tuple with the i-th bounding box and associated confidence score. If the detections in $\mathcal{B}$ ensures that $\mathrm{IoU}(\hat{b}_i, b_k) > \mathrm{IoU}(\hat{b}_j, b_k)$ if $\hat{p}_i > \hat{p}_j$ for all $i \neq j$ and $k$ is the ground truth that each detection has maximum IoU with, then $\mathcal{B}$ provides the optimal AP for class c given $\mathcal{B}^{raw}$. The value of the optimal AP in this case is $\frac{\mathrm{N}_{TP}(\mathcal{B}^{raw})}{M}$, where $\mathrm{N}_{TP}(\mathcal{B}^{raw})$ is the number of TPs in $\mathcal{B}^{raw}$ and $M > 0$ is the number of ground-truth objects from class c.*

*Proof.* Below we show that $\mathcal{B}$ satisfies all the three necessary conditions required to maximize the AP for any IoU threshold $\tau$ (to identify TPs):

1. *$\mathcal{B}$ is to have $\mathrm{N}_{TP}(\mathcal{B}^{raw})$, that is, no TP is to be removed in $\mathcal{B}^{raw}$ by postprocessing.* Note that this is handled by post-processing as an assumption in the Lemma.
2. *The minimum confidence score of TPs in $\mathcal{B}$ is to be higher than the maximum confidence score of FPs.* This is also ensured considering (i) the assumption that is $\mathrm{IoU}(\hat{b}_i, b_k) > \mathrm{IoU}(\hat{b}_j, b_k)$ if $\hat{p}_i > \hat{p}_j$ for all $i \neq j$ and $k$ is the ground truth that each detection has maximum IoU with; and (ii) the duplicates are removed by NMS. As a result, all TPs are ranked higher than all FPs.
3. *$\mathcal{B}$ is to include the detection with the largest IoU with each ground truth $k$.*[*]. NMS selects the detection with the highest confidence score among a group of overlapping detections. Considering that NMS does not remove any TPs and $\mathrm{IoU}(\hat{b}_i, b_k) > \mathrm{IoU}(\hat{b}_j, b_k)$ for all $i$ and $j$ if $\hat{p}_i > \hat{p}_j$ and $i \neq j$ holds, NMS survives the detections with the best localisation quality as they have the highest scores in their groups. As a result, this condition is also satisfied.

To compute the area under the precision-recall curve, we need the precision and recall pairs. Please note that once the aforementioned criteria are satisfied, the precision will be 1 when recall interval is between $[0, \frac{\mathrm{N}_{TP}(\mathcal{B}^{raw})}{M}]$; and will be zero beyond this. Therefore, the area under the precision-recall curve trivially turns out to be $\frac{\mathrm{N}_{TP}(\mathcal{B}^{raw})}{M}$. □

**On the Optimality of Eq. 4**  Based on Lemma 1, we now present our theorem and its proof.

**Theorem 1.** *Assume that E different experts are combined in the form of an MoE and the detection set of e-th expert for class c is denoted by $\mathcal{B}_e$. Given the union of these detections (before aggregation) is denoted by $\mathcal{B}^{raw} = \bigcup_{e=1}^{E} \mathcal{B}_e$, the MoE using perfectly calibrated experts in terms of Eq. 4 yields optimal AP for class c over any possible MoEs following Lemma 1. Accordingly, denoting the number of TPs in $\mathcal{B}^{raw}$ by $\mathrm{N}_{TP}(\mathcal{B}^{raw})$ and the number of ground-truth objects for class c by $M > 0$, the resulting AP is $\frac{\mathrm{N}_{TP}(\mathcal{B}^{raw})}{M}$.*

*Proof.* Please note that it is trivial to show that the detections in $\mathcal{B}$ of the perfectly calibrated detectors ensures the requirement in Lemma 1 that $\mathrm{IoU}(\hat{b}_i, b_k) > \mathrm{IoU}(\hat{b}_j, b_k)$ if $\hat{p}_i > \hat{p}_j$ for all $i \neq j$ and $k$ is the ground truth that each detection has maximum IoU with. This is because the calibration target ensures that $\hat{p}_i = \mathrm{IoU}(\hat{b}_i, b_k)$ for each detection $i$. As a result, following from Lemma 1, the theorem holds. □

Please note that it is trivial to show that Theorem 1 generalizes to the cases in which the dataset involves multiple classes and COCO-style AP is used. This is because the former case is estimated as the average of the APs over different classes and the latter is simply the average of the APs over different IoU thresholds (please see footnote * for further discussion). Consequently, as the class-wise APs are optimal, so do the dataset AP and the COCO-style AP. Please refer to App. D.7 for the experiment presenting the effect of Theorem 1 using an Oracle MoE.

---

[*] When we consider the AP for a single $\tau$, this criteria is not mandatory to be satisfied as the conventional AP consider localisation performance loosely(Oksuz et al., 2021c). However, the localisation performance is an important aspect of an object detector and it is considered by various performance measures such as COCO-style AP, which corresponds to the average over APs with 10 different $\tau$ thresholds or LRP Error(Oksuz et al., 2021c). As a result, while we consider the AP for a single $\tau$ in this section, in order for our theoretical justifications to be applicable to other performance measures, we also take this criterion in account.

# D   Further Experiments and Analyses

Here, we present further experiments and analyses that are not included in the main text due to space limitation.

## D.1   Further Details on Used Models

We provide the details of the used models as follows. We again note that we haven't trained any model but used off-the-shelf detectors with the exception of DEs. Here we provide further details on the used detectors. Still, as it is not feasible to provide all of the details, we also present the papers and repositories that we borrow these off-the-shelf models in order to ensure the reproducibility of our results.

**Object Detection**   We use two different configurations. In the first one, we employ three detectors with ResNet-50 (He et al., 2016) with FPN (Lin et al., 2017a) backbone. These detectors are:

- Rank & Sort R-CNN (RS R-CNN) (Oksuz et al., 2021a) is a recent representative of the two-stage R-CNN family (Ren et al., 2017; Dai et al., 2016; Zhang et al., 2020a) optimizing a ranking-based loss function,
- Adaptive Training Sample Selection (ATSS) (Zhang et al., 2020b) is a common one stage baseline,
- Probabilistic Anchor Assignment (PAA) (Kim & Lee, 2020) relies on the one-stage ATSS architecture but with a different anchor assignment mechanism and postprocessing of the confidence scores.

We obtain RS R-CNN and ATSS from (Oksuz et al., 2023) and PAA from (Chen et al., 2019). All these detectors are trained for 36 epochs using multi-scale training data augmentation in which the shorter side of the image is resized within the range of [480, 800] for RS R-CNN and ATSS and [640, 800] for PAA. We do not use Soft NMS and Score Voting for the single detectors.

In our second setting, we use the following detectors:

- YOLOv7 (Wang et al., 2022a) with a large convolutional backbone following its original setting,
- QueryInst (Fang et al., 2021) as a transformer-based detector with a Swin-L (Liu et al., 2021) backbone,
- ATSS with transformer-based dynamic head (Dai et al., 2021) and again Swin-L backbone.

Again, we obtain YOLOv7 and dynamic head from mmdetection (Chen et al., 2019) and use the official repository of QueryInst (Fang et al., 2021).

Furthermore, to improve the SOTA on COCO *test-dev*, we use two of the most recent and strong public detectors, EVA (Fang et al., 2023) and Co-DETR (Zong et al., 2023). EVA (Fang et al., 2023) is a foundation model for computer vision that utilises Cascade Mask R-CNN (Cai & Vasconcelos, 2018) to perform object detection wheras Co-DETR (Zong et al., 2023) is a recent transformer-based detector. For both of them, we directly consider the official repositories and do not change any settings, including the Soft-NMS for EVA (Fang et al., 2023).

**Rotated Object Detection**   For rotated object detection, we use RTMDet and LSKN as two different detectors. We obtain RTMDet again from mmdetection (which is also the official repository for RTMDet) and LSKN from its official repository (Li et al., 2023).

**Open-Vocabulary Object Detection**   We use two of the most recent works on vision-language foundation models literature, which can be listed as follows:

- Grounding DINO (Liu et al., 2024) is a transformer-based detector.
- MQ-GLIP from the recent (Xu et al., 2023), an anchor-based detector that introduces multi-modal queries on top of GLIP (Li et al., 2022b).

For both Grounding Dino (Liu et al., 2024) and MQ-GLIP (Xu et al., 2023), we consider the versions employing Swin-T (Liu et al., 2021) as the backbone. We do not perform any prompt engineering and do not change any settings. Furthermore, we directly utilise the official GitHub repositories for both of the models.

Table A.15: Accuracy and calibration performance of uncalibrated, early calibrated and late calibrated models on COCO *mini-test.* A red cell indicates a notable AP drop compared to the uncalibrated detector while a green cell implies consistency. Please note that the calibration performance of early and late calibration should not be compared as they differ in their train and test sets (pre-NMS vs. post-NMS). Still for the interested reader, an example is that early calibration yields 8.3, 17.1 and 9.7 LaECE for ATSS, PAA and RS R-CNN respectively once evaluated on post-NMS detections. CW: Class-wise, CA: class-agnostic, bold: best calibration performance, underlined: second best. Calibrator is not available (N/A) for uncalibrated models. CA IR provides a good balance of AP and calibration performance. The results are presented on COCO *minitest.*

| Cal. Type | Class Type | Calibrator | AP | | | LaECE | | | LaACE | | | LaMCE | | |
|---|---|---|---|---|---|---|---|---|---|---|---|---|---|---|
| | | | RS R-CNN | ATSS | PAA | RS R-CNN | ATSS | PAA | RS R-CNN | ATSS | PAA | RS R-CNN | ATSS | PAA |
| | N/A | N/A | 42.4 | 43.1 | 43.2 | 13.79 | 0.20 | 3.39 | **5.18** | 25.22 | 9.73 | 35.37 | 42.53 | 19.80 |
| Early Cal. | CW | LR | 26.4 | 42.0 | 37.1 | 0.44 | 0.12 | 0.22 | 25.12 | 10.13 | 28.95 | 61.11 | 24.79 | 60.05 |
| | | IR | 41.9 | 42.8 | 42.5 | **0.03** | **0.02** | 1.39 | 5.23 | **5.40** | **5.26** | 25.10 | 25.38 | 26.60 |
| | CA | LR | 42.4 | 42.8 | 43.2 | 0.65 | 0.17 | 0.37 | 27.48 | 7.86 | 25.49 | 68.81 | 23.07 | 54.03 |
| | | IR | 42.4 | 43.1 | 43.2 | 0.14 | 0.10 | **0.14** | 5.86 | 6.43 | 6.70 | **14.08** | 15.59 | 18.15 |
| | N/A | N/A | 42.4 | 43.1 | 43.2 | 36.45 | 5.01 | 11.23 | 29.30 | 17.48 | 15.79 | 45.42 | 40.00 | 32.33 |
| Late Cal. | CW | LR | 42.4 | 43.1 | 43.2 | 4.36 | 2.69 | 1.39 | 14.19 | **9.03** | 12.01 | 40.20 | **29.51** | 33.76 |
| | | IR | 41.8 | 42.5 | 42.4 | **1.56** | 2.35 | **1.21** | 9.21 | 9.81 | **9.24** | 38.43 | 40.34 | 37.84 |
| | CA | LR | 42.4 | 43.0 | 43.2 | 5.83 | 4.46 | 1.63 | 13.86 | 9.46 | 11.88 | 37.79 | 29.59 | **29.91** |
| | | IR | 42.3 | 43.1 | 43.2 | 3.15 | 4.51 | 1.62 | **8.93** | 9.51 | 9.61 | **35.72** | 37.35 | 35.59 |

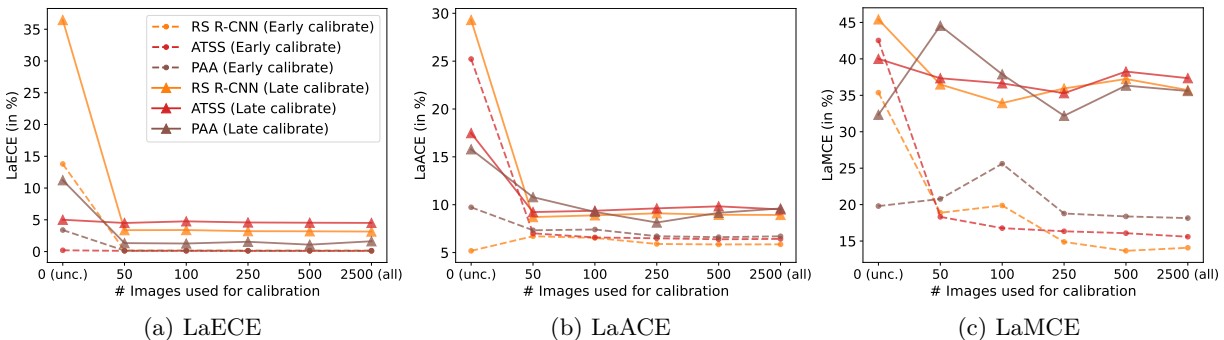

(a) LaECE  (b) LaACE  (c) LaMCE

Figure A.6: The effect of number of images on calibration using CA IR. We find it sufficient to use only 500 images for calibration. The results are presented using COCO *minitest.*

**Instance Segmentation** We use three different Mask R-CNN variants for instance segmentation:

- The vanilla Mask R-CNN (He et al., 2017) with ResNeXt-101 (Xie et al., 2016) backbone, softmax classifier and using Repeat Factor Sampling (RFS) to address the long-tailed nature of LVIS,
- Mask R-CNN with ResNet-50, sigmoid classifier, trained with RS Loss (Oksuz et al., 2021a) and RFS,
- Mask R-CNN with ResNet-50, softmax classifier, trained with Seesaw Loss(Wang et al., 2020) but no RFS.

We obtain Vanilla Mask R-CNN and Seesaw Loss from mmdetection. As for Mask R-CNN trained with RS Loss, we use the official repository of RS Loss (Oksuz et al., 2021a) in which it is trained for 12 epochs using multi-scale training augmentation. The other Mask R-CNN variants also employ multi-scale training augmentation and the Vanilla Mask R-CNN is trained for 12 epochs as well. Differently, Mask R-CNN with Seesaw Loss is trained for 24 epochs and uses the mask normalization technique proposed in the same paper(Wang et al., 2020).

### D.2 Further Ablation Experiments on MoCaE

This section presents further ablation experiments by which we validate our design choices. These include the validation of calibration method and the design choices in Refining NMS.

### D.2.1 Further Details and Ablation on Calibration

We use CA IR and LR to calibrate the models given the final scores in MoCaE, i.e., late calibration. More specifically, we use CA IR in all of the experiments unless otherwise explicitly specified. One notable exception where we use the CA LR instead of IR is the SOTA experiment in Tab. 8 in which we observe that CA LR performs 0.1 AP compared to CA IR. Please note that, we obtain those calibrators on a held-out validation set consisting of only 500 images. This section presents further experiments to validate these design choices.

**Validating the Calibrator** While calibrating the predictive distribution of the classifiers, commonly the accuracy of the classifier is preserved. However, this might not be the case once the predictive confidence instead of the distribution is calibrated, which is the case in the common calibrators for object detection and in our case (Oksuz et al., 2023). This is mainly because the ranking among the detections can change unless the calibrator is a monotonically increasing function of the confidence. Therefore, we would ideally expect a calibrator to improve the calibration by at least preserving the accuracy of the detector. To ensure that, we investigate LR and IR both CA and Class-wise (CW) on late as well as early calibration in Tab. A.15. Here, we obtain the calibrators on COCO *minival* and report the calibration error and accuracy on COCO *minitest*. We observe in the red cells in Tab. A.15 that all calibrators, except CA IR and CA IR, decrease AP especially for early calibration. Furthermore, these calibrators improve LaECE in all cases. These observations on accuracy and calibration led us to choose CA IR and CA LR while calibrating the single models in MoCaE.

**500 Images are Sufficient for calibration in MoCaE** Equipped with the aforementioned insights, we then investigate the sufficient number of images for calibration to enable MoEs using MoCaE. Our aim here is to determine the cardinality of the held-out validation set to properly calibrate the models, which can help the practitioners to avoid reserving redundant data for this held-out validation set. Following the literature (Guo et al., 2017; Oksuz et al., 2023), we obtain calibrators on a held-out validation set (COCO minival) and then report the results on the test set (COCO minitest). Furthermore, we obtain several calibrators on both early and late calibration settings by using different number of images. Fig. A.6 presents how LaECE, LaACE and LaMCE change when the cardinality of the hold-out validation set changes on three different detectors. We observe in general that calibration errors drop significantly even when only 50 images are used to learn the calibrators. Through introducing more images, while we do not observe a notable gain in LaECE (Fig. A.6(a)), LaACE (Fig. A.6(b)) and LaMCE (Fig. A.6(c)) continue to improve especially for early calibration, implying the necessity for these calibration errors. Overall, as we have not observed a notable gain after 500 images, we keep 500 images on the held-out validation set while training the calibrators. A noteworthy point is that 500 images is a small number images compared to large training sets in object detection, demonstrating the ease of applicability of using MoCaE.

**Comparison of Early and Late Calibration** Here, we provide additional insights on early and late calibration.In Tab. A.16, we can see that while both approaches improve single models, late calibration performs slightly better than early calibration consistently. Furthermore, in practical terms, late calibration is significantly simpler as the number of confidence scores obtained before post-processing is significantly larger than those obtained after postprocessing (i.e., final detections). To illustrate this more concretely, RS R-CNN outputs 1K proposals for each image as raw detections, thus resulting with a single image containing 80K raw confidence scores for the COCO dataset and more than 1M scores for LVIS as each proposal has a score for each class. In addition, a very large amount of these raw detections do not even overlap with any objects, complicating the problem due to this imbalanced nature of the data. Furthermore, the number of raw confidence scores is significantly larger for one-stage detectors (ATSS and PAA in our case) as such detectors make predictions directly from a very large number of anchors; making early calibration even more

Table A.16: Further experiments comparing early and late calibration. Standard NMS is used for both of the methods. While both approaches improve single models, late calibration performs slightly better. The results are presented on COCO *minitest*.

| Calibration Type | RS R-CNN | ATSS | PAA | AP | $AP_{50}$ | $AP_{75}$ | $AP_S$ | $AP_M$ | $AP_L$ |
|---|---|---|---|---|---|---|---|---|---|
| N/A | ✓ | | | 42.4 | 62.1 | 46.2 | 26.8 | 46.3 | 56.9 |
| (Single | | ✓ | | 43.1 | 61.5 | 47.1 | 27.8 | 47.5 | 54.2 |
| Models) | | | ✓ | 43.2 | 60.8 | 47.1 | 27.0 | 47.0 | 57.6 |
| Early | ✓ | ✓ | | 43.9 | 63.5 | 47.9 | 28.6 | 47.9 | 56.9 |
| Late | ✓ | ✓ | | 44.1 | 63.0 | 48.4 | 28.5 | 48.4 | 56.8 |
| Early | ✓ | | ✓ | 43.8 | 63.3 | 47.7 | 28.1 | 47.8 | 57.6 |
| Late | ✓ | | ✓ | 44.0 | 62.7 | 47.9 | 28.2 | 48.2 | 58.1 |
| Early | | ✓ | ✓ | 44.3 | 63.1 | 47.8 | 28.8 | 48.1 | 56.7 |
| Late | | ✓ | ✓ | 44.4 | 62.5 | 48.5 | 29.2 | 48.5 | 57.3 |
| Early | ✓ | ✓ | ✓ | 44.5 | **63.7** | 48.4 | 29.0 | 48.5 | 57.5 |
| Late | ✓ | ✓ | ✓ | **44.7** | 63.1 | **48.9** | **29.2** | **49.0** | **58.2** |

Table A.17: Comparison of different calibration methods to obtain MoEs. CA IR performs better than other methods. The results are presented on COCO *minitest*.

| Calibrated | RS R-CNN | ATSS | PAA | AP | AP50 | AP75 | APS | APM | APL |
|---|---|---|---|---|---|---|---|---|---|
| N/A | ✓ | | | 42.4 | 62.1 | 46.2 | 26.8 | 46.3 | 56.9 |
| (Single | | ✓ | | 43.1 | 61.5 | 47.1 | 27.8 | 47.5 | 54.2 |
| Models) | | | ✓ | 43.2 | 60.8 | 47.1 | 27.0 | 47.0 | 57.6 |
| ✗ | ✓ | ✓ | | 42.4 | 62.1 | 46.3 | 26.8 | 46.3 | 56.9 |
| CW LR | ✓ | ✓ | | 43.0 | 61.4 | 46.9 | 28.1 | 47.2 | 54.4 |
| CA LR | ✓ | ✓ | | 43.7 | 62.5 | 47.7 | 28.7 | 48.2 | 55.2 |
| CA IR | ✓ | ✓ | | 44.1 | 63.0 | 48.4 | 28.5 | 48.4 | 56.8 |
| ✗ | ✓ | | ✓ | 43.4 | 62.5 | 47.1 | 27.3 | 47.3 | 58.0 |
| CW LR | ✓ | | ✓ | 42.7 | 61.8 | 46.7 | 27.1 | 46.5 | 57.6 |
| CA LR | ✓ | | ✓ | 43.5 | 63.0 | 47.7 | 27.9 | 47.5 | 57.8 |
| CA IR | ✓ | | ✓ | 44.0 | 62.7 | 47.9 | 28.2 | 48.2 | 58.1 |
| ✗ | | ✓ | ✓ | 43.3 | 60.9 | 47.2 | 27.1 | 47.2 | 57.6 |
| CW LR | | ✓ | ✓ | 42.6 | 60.7 | 46.5 | 26.6 | 47.2 | 53.9 |
| CA LR | | ✓ | ✓ | 44.4 | 62.5 | 48.3 | 29.0 | 48.3 | 57.3 |
| CA IR | | ✓ | ✓ | 44.4 | 62.5 | 48.5 | 29.2 | 48.5 | 57.3 |
| ✗ | ✓ | ✓ | ✓ | 43.4 | 62.5 | 47.1 | 27.3 | 47.3 | 58.0 |
| CW LR | ✓ | ✓ | ✓ | 43.0 | 61.3 | 47.1 | 28.2 | 47.4 | 54.6 |
| CA LR | ✓ | ✓ | ✓ | 44.0 | 62.7 | 48.1 | 29.0 | 48.5 | 55.8 |
| CA IR | ✓ | ✓ | ✓ | 44.7 | 63.1 | 48.9 | 29.2 | 49.0 | 58.2 |

impractical for them[*]. On the other hand, we use only top-100 detections in COCO and top-300 detections in LVIS for each image following the evaluation specification of these datasets. Thereby resulting in more practical scenarios with significantly smaller number of detections for late calibration compared to early. Consequently, considering its slight accuracy gain as well as simplicity, we prefer late calibration over early to obtain MoEs in MoCaE.

---

[*]To keep this number manageable, we use top-1000 detections predicted from each pyramid level for ATSS and PAA for early calibration.

Table A.18: Ablation analysis of MoCaE. We use the MoE combining RS R-CNN, ATSS and PAA on COCO *mini-test*

| MoCaE Pipeline | Calibration | Refining NMS | | AP |
| --- | --- | --- | --- | --- |
| | | Soft NMS | Score Voting | |
| Early | ✗ | ✗ | ✗ | 43.3 |
| Early | ✓ | ✗ | ✗ | 44.5 |
| Late | ✓ | ✗ | ✗ | 44.7 |
| Late | ✓ | ✓ | ✗ | 44.8 |
| Late | ✗ | ✓ | ✗ | 43.4 |
| Late | ✗ | ✓ | ✓ | 44.4 |
| Late | ✓ | ✓ | ✓ | **45.5** |

Table A.19: Using a subset of the training set for calibrating object detectors using MoCaE. The results are reported on COCO *minitest*. We compare the models in Tab. 6.

| Model Type | Detector | AP | AP$_{50}$ | AP$_{75}$ |
| --- | --- | --- | --- | --- |
| Single Models | RS R-CNN | 42.4 | 62.1 | 46.2 |
| | ATSS | 43.1 | 61.5 | 47.1 |
| | PAA | 43.2 | 60.8 | 47.1 |
| Deep Ensembles | RS R-CNN $\times$ 5 | 43.4 | 63.0 | 47.7 |
| | ATSS $\times$ 5 | 44.1 | 62.3 | 48.4 |
| | PAA $\times$ 5 | 44.4 | 62.0 | 48.4 |
| Mixtures of Experts | Vanilla MoE (RS R-CNN, ATSS, PAA) | 43.4 | 62.5 | 47.1 |
| | MoCaE (ATSS and PAA) calibration on *train* | 44.7 | 62.3 | 49.0 |
| | MoCaE (ATSS and PAA) calibration on *val* | 44.8 | 62.4 | 49.2 |
| | MoCaE (RS R-CNN, ATSS, PAA) calibration on *train* | 45.4 | 63.0 | **50.0** |
| | MoCaE (RS R-CNN, ATSS, PAA) calibration on *val* | **45.5** | **63.2** | **50.0** |

**Effect of Different Calibration Methods on MoEs**   While we choose CA IR as our calibration method in MoCaE, here we present how different calibration methods perform in obtaining MoEs. Specifically, we use late calibration with CA LR and CW LR as these two methods also preserve the accuracy of single models as shown in Tab. A.17. Tab. A.17 presents the results where we can see that CA calibrators perform better than CW LR. Also, while CA LR obtains on par performance with CA IR while combining ATSS and PAA, it performs worse once RS R-CNN is in the mixture. This might be because the calibration error of CA LR is higher than CA IR (for late calibration) in terms of all calibration measures as shown in Tab. A.17).

**Ablation of MoCaE with Early Calibration**   Tab. A.18 presents a more detailed version of the Tab. 14 included in the paper. In this version, we also include early calibration, which performs similar with late calibration as shown in Tab. A.18.

**Can we use training set to calibrate experts in MoCaE?**   Following the common convention in the classification literature (Mukhoti et al., 2020; Guo et al., 2017) and the recently-proposed approaches in object detection (Kuzucu et al., 2024; Oksuz et al., 2023), we obtain the post-hoc calibrators on a held-out validation set while using MoCaE. In our experiments, this held-out validation set includes 500 images. While this is usually a quite small dataset to keep for calibration, we still investigate if a subset of the training set can be used for the purpose of combining detectors. To do so, we randomly sample 500 images from the training set of COCO and LVIS, and report the results in Tab. A.19 and Tab. A.20. The main observation in these tables is that *while calibration on a subset of the training set and on a validation set perform almost on-par on COCO, there is a significant performance gap (1.0 AP) in favor of using validation set for LVIS.* Please note that, LVIS is a long-tailed dataset with around 1K different classes and it has been shown in the literature that the models tend to overfit for the datasets with long-tailed examples (Feldman & Zhang, 2020;

Table A.20: Using a subset of the training set for calibrating object detectors using MoCaE. The results are reported on LVIS *val* set (excluding 500 images used for calibration). We compare the models in Tab. 11.

| Method | AP | $AP_{50}$ | $AP_{75}$ | $AP_r$ | $AP_c$ | $AP_f$ |
|---|---|---|---|---|---|---|
| Mask R-CNN | 25.4 | 39.2 | 27.3 | 15.7 | 24.7 | 30.4 |
| RS Loss | 25.1 | 38.2 | 26.8 | 16.5 | 24.3 | 29.9 |
| Seesaw Loss | 25.4 | 39.5 | 26.9 | 15.8 | 24.7 | 30.4 |
| Vanilla MoE | 25.2 | 38.3 | 26.8 | 16.5 | 24.3 | 29.9 |
| MoCaE calibration on *train* | 26.7 | 40.7 | 28.6 | 17.6 | 25.9 | 31.4 |
| MoCaE calibration on *val* | **27.7** | **42.8** | **29.4** | **18.2** | **27.3** | **32.4** |

Table A.21: Sensitivity of Vanilla MoE and MoCaE to different configurations of Soft NMS. The results are presented on COCO mini-test and 500 validation images that we used to train the calibrators for LVIS. We report box AP for COCO and mask AP for LVIS.

| Method | Soft NMS | COCO | LVIS |
|---|---|---|---|
| Vanilla MoE | ✗ | **43.4** | 37.5 |
| | Linear, $IoU_{NMS} = 0.65$ | **43.4** | 37.5 |
| | Gaussian, $\sigma_{NMS} = 0.20$ | 41.6 | **37.9** |
| | Gaussian, $\sigma_{NMS} = 0.40$ | 42.1 | 37.8 |
| | Gaussian, $\sigma_{NMS} = 0.60$ | 42.4 | **37.9** |
| | Gaussian, $\sigma_{NMS} = 0.80$ | 42.7 | 37.8 |
| | Gaussian, $\sigma_{NMS} = 1.00$ | 42.9 | 37.9 |
| MoCaE | ✗ | 44.7 | 39.8 |
| | Linear, $IoU_{NMS} = 0.65$ | **44.8** | 39.9 |
| | Gaussian, $\sigma_{NMS} = 0.20$ | 43.7 | 40.6 |
| | Gaussian, $\sigma_{NMS} = 0.40$ | 44.4 | **40.8** |
| | Gaussian, $\sigma_{NMS} = 0.60$ | **44.8** | 40.6 |
| | Gaussian, $\sigma_{NMS} = 0.80$ | 44.7 | 40.2 |
| | Gaussian, $\sigma_{NMS} = 1.00$ | 44.6 | 39.9 |

Feldman, 2019), implying a larger generalisation gap (i.e., the difference between the test and the training errors) for such datasets. Subsequently, Mukhoti et al. (2020)and Bai et al. (2022) showed that the larger generalisation gap implies a larger calibration error. Accordingly, the results in Tab. A.20 shows the resulting drawback of using training sets for calibration compared to using a held-out validation set. Specifically, calibrating on validation set outperforms calibrating on a subset of the training set by 1.0 mask AP. We also note that, calibrating on the training set still outperforms Vanilla MoE by 1.5 mask AP, which demonstrates the importance of the calibration in obtaining an MoE.

### D.2.2  Sensitivity of MoCaE to Design Choices in Refining NMS

Refining NMS combines Soft NMS and Score Voting. Specifically, Soft NMS can be linear or gaussian; furthermore both Soft NMS (either linear or gaussian) and Score Voting have hyper-parameters. Here, we investigate the sensitivity of MoCaE to such design choices using Soft NMS as an example using RS R-CNN, ATSS and PAA for COCO; and the setting described in Sec. 4 for LVIS. We can easily see in Tab. A.21 that Vanilla MoE does not benefit properly from Soft NMS without calibration. For example, there is no gain for Linear Soft NMS, the performance degrades for the Gaussian Soft NMS on COCO and the gain is only 0.4 for LVIS. This is expected as a single hyper-parameter to reconciliate the scores all detectors might not be sufficient especially for the Gaussian Soft NMS. On the other hand, after calibration, we consistently see the gains for our MoCaE: MoCaE benefits slightly on COCO dataset both for linear and gaussian cases; and besides, the gain on LVIS is 1.0 mask AP. This is because, the scores are compatible for each detector

Table A.22: Object detection performance on COCO *minitest* with the detectors using multiscale testing. Test image scale refers to the shorter side of the image, please refer to the text for the details. MoCaE is complementary to multiscale testing. Our gains in green are obtained compared to the best single model for each performance measure, represented as underlined.

| Model Type | Detector | Test Image Scale | AP | $AP_{50}$ | $AP_{75}$ | $AP_S$ | $AP_M$ | $AP_L$ |
|---|---|---|---|---|---|---|---|---|
| Single Models | RS R-CNN | 800 | 42.4 | 62.1 | 46.2 | 26.8 | 46.3 | 56.9 |
| | ATSS | 800 | 43.1 | 61.5 | 47.1 | 27.8 | 47.5 | 54.2 |
| | PAA | 800 | 43.2 | 60.8 | 47.1 | 27.0 | 47.0 | 57.6 |
| | RS R-CNN | 400, 800, 1200 | 43.2 | 62.1 | 47.6 | 29.4 | 46.5 | 58.8 |
| | ATSS | 400, 800, 1200 | 44.7 | 62.6 | 49.0 | 31.6 | 48.2 | 57.5 |
| | PAA | 400, 800, 1200 | 44.7 | 62.0 | 49.0 | 31.6 | 48.2 | 58.5 |
| Mixtures of Experts | Vanilla MoE (RS R-CNN, ATSS, PAA) - Ours | 800 | 43.4 | 62.5 | 47.1 | 27.3 | 47.3 | 58.0 |
| | MoCaE (RS R-CNN, ATSS, PAA) - Ours | 800 | 45.5 | 63.2 | 50.0 | 29.7 | 49.7 | 59.3 |
| | Vanilla MoE (RS R-CNN, ATSS, PAA) | 400, 800, 1200 | 44.4 | 62.5 | 48.7 | 30.2 | 47.8 | 59.6 |
| | MoCaE (RS R-CNN, ATSS, PAA) - Ours | 400, 800, 1200 | **46.4** | **63.3** | **51.6** | **32.9** | **50.0** | **61.2** |
| | | | +1.8 | +0.7 | +2.6 | +1.3 | +1.8 | +2.7 |

after calibration and a single hyperparameter allows Soft NMS to properly adjust the scores from different detectors. We choose Linear Soft NMS on COCO resulting in the best results for Vanilla MoE and MoCaE. For LVIS, we use Gaussian Soft NMS with $\sigma_{NMS} = 0.40$ for MoCaE. In a similar way, we validate the hyper-parameter of Score Voting as 0.04.

### D.2.3  MoCaE is Complementary to Multiscale Testing

Multiscale test augmentation is another way of improving the detection performance of a detector, which has been commonly used in the literature (Oksuz et al., 2020; Zhang et al., 2020b; Liu et al., 2018; Duan et al., 2019). In this section, we show that MoCaE is complementary to multiscale testing. To show this, we first apply multi-scale testing to RS R-CNN, ATSS and PAA as the individual detectors. Specifically, while we use only images with 800x1333 resolution for inference, here we use images with 400x667, 800x1333 and 1200x2000 resolutions as three different scales. Tab. A.22 shows that this way of testing improves the performance of the single models between 0.8 to 1.6 AP. Then, we combine the outputs of these models with multiscale augmentation using Vanilla MoE, and MoCaE and make two more important observations in the same Tab. A.22: (i) The performance of MoCaE with multi-scale detectors improves by 0.9 AP compared to the MoCaE with detectors without test time augmentation; and (ii) MoCaE improves the single best detector with multiscale testing by 1.8AP. These observations lead to the conclusion that MoCaE is complementary to multiscale testing.

### D.3  Further Details and Analyses on Deep Ensembles

This section provides further details and analyses on DEs.

### D.3.1  The Effect of Calibration on DEs

DEs combine the same models that are trained from different initialization of the parameters. Ideally, the expectation over the predictive distributions of the components in a DE yields the prediction of the DE. This can be easily obtained for classifiers which predict a categorical distribution over the classes given an input image. On the other hand, it is not straightforward to use DEs for detectors as there is no clear way to associate detections from different detectors. As a result, similar to MoCaE, we obtain DEs by using late calibration as shown in Fig. 4(a), which turns out to be an effective method. To see that, we first present the single model performance of the five different components comprising DEs in Tab. A.23. Then, from these single detectors, we obtain DEs for PAA with and without calibration using the standard NMS. Tab. A.24 shows that this way of obtaining DEs is effective as the performance increases when the number of components increases. We observe that increasing the number of components improve the performance

Table A.23: Single model performance of the detectors that we used in DEs. While obtaining MoEs, we combine "Model 1" of different types of detectors.

| Model | AP | $AP_{50}$ | $AP_{75}$ | $AP_S$ | $AP_M$ | $AP_L$ |
|---|---|---|---|---|---|---|
| RS R-CNN (Model 1) | 42.4 | 62.1 | 46.2 | 26.8 | 46.3 | 56.9 |
| RS R-CNN (Model 2) | 42.6 | 62.7 | 46.7 | 27.3 | 46.5 | 55.8 |
| RS R-CNN (Model 3) | 42.6 | 62.6 | 46.5 | 27.2 | 46.8 | 56.1 |
| RS R-CNN (Model 4) | 42.6 | 62.3 | 46.1 | 27.9 | 46.1 | 55.9 |
| RS R-CNN (Model 5) | 42.2 | 62.8 | 45.8 | 26.7 | 46.5 | 55.7 |
| ATSS (Model 1) | 43.1 | 61.5 | 47.1 | 27.8 | 47.5 | 54.2 |
| ATSS (Model 2) | 43.3 | 61.5 | 47.5 | 28.9 | 47.8 | 55.0 |
| ATSS (Model 3) | 43.3 | 61.5 | 47.0 | 28.9 | 47.9 | 55.8 |
| ATSS (Model 4) | 43.0 | 61.2 | 46.8 | 29.0 | 47.4 | 54.6 |
| ATSS (Model 5) | 43.3 | 61.5 | 47.5 | 28.2 | 47.7 | 54.8 |
| PAA (Model 1) | 43.2 | 60.8 | 47.1 | 27.0 | 47.0 | 57.6 |
| PAA (Model 2) | 43.5 | 61.0 | 47.3 | 27.5 | 47.7 | 57.7 |
| PAA (Model 3) | 43.4 | 61.1 | 47.2 | 27.7 | 47.6 | 57.8 |
| PAA (Model 4) | 43.6 | 61.4 | 47.2 | 27.6 | 47.6 | 57.6 |
| PAA (Model 5) | 43.6 | 61.3 | 47.2 | 27.9 | 47.9 | 58.0 |

Table A.24: The effect of increasing the components and using calibration on DEs. Increasing the components improves the performance while calibration does not have a notable effect on performance for DEs unlike their importance for MoEs.

| Model | Calibration | AP | $AP_{50}$ | $AP_{75}$ | $AP_S$ | $AP_M$ | $AP_L$ |
|---|---|---|---|---|---|---|---|
| PAA $\times$ 2 | ✗ | 43.8 | 61.4 | 47.6 | 28.2 | 48.3 | 58.3 |
| PAA $\times$ 2 | ✓ | 43.9 | 61.4 | 47.7 | 28.0 | 48.3 | 58.6 |
| PAA $\times$ 3 | ✗ | 44.1 | 61.5 | 48.0 | 28.8 | 48.7 | 58.8 |
| PAA $\times$ 3 | ✓ | 44.1 | 61.6 | 48.0 | 28.7 | 48.7 | 58.9 |
| PAA $\times$ 4 | ✗ | 44.3 | 61.7 | 48.3 | 28.9 | 48.8 | 59.1 |
| PAA $\times$ 4 | ✓ | 44.3 | 61.7 | 48.3 | 28.9 | 48.8 | 59.1 |
| PAA $\times$ 5 | ✗ | 44.4 | 61.7 | 48.6 | 29.1 | 49.0 | 59.3 |
| PAA $\times$ 5 | ✓ | 44.4 | 61.7 | 48.6 | 28.9 | 49.0 | 59.3 |

between $0.1 - 0.3$ AP. On the other hand, as there is no incompatibility among different detectors in a DE, the effect of calibration is not notable for DEs. Still, having observed that PAA with 2 components has a slightly better (0.1 AP) performance once calibrated, in our comparisons we use DEs with calibration.

### D.3.2    DEs with Less Components and Refining NMS

In Tab. 6, we compared MoCaE with DEs with 5 components. Note that in this case, the DEs have more components, implying a higher number of parameters compared to our MoCaE with 2 or 3 components. For the sake of completeness and provide a more fair comparison to our MoCaE with a maximum of 3 components , Tab. A.25 extends our comparison in Tab. 6 by including (i) the DEs with 3 components and (ii) the DEs with Refining NMS. Tab. A.25 presents that with equal number of components, our MoCaE outperforms the best DE with 3 components by 1.4 AP (44.1 of PAA $\times$3 vs 45.5 AP of MoCaE). Furthermore, in the case that Refining NMS is used for MoCaEs, their performance consistenty improves; showing the effectiveness of our aggregator. Still, the performance of the best DE, i.e., PAA with 5 components and Refining NMS, is 0.5 AP lower compared to our MoCaE; demonstrating the effectiveness of our approach.

Table A.25: The effect of Refining NMS on Deep Ensembles and Vanilla MoE.

| Model Type | Detector | AP | AP$_{50}$ | AP$_{75}$ | AP$_S$ | AP$_M$ | AP$_L$ |
|---|---|---|---|---|---|---|---|
| Single Models | RS R-CNN | 42.4 | 62.1 | 46.2 | 26.8 | 46.3 | 56.9 |
| | ATSS | 43.1 | 61.5 | 47.1 | 27.8 | 47.5 | 54.2 |
| | PAA | 43.2 | 60.8 | 47.1 | 27.0 | 47.0 | 57.6 |
| Deep Ensembles | RS R-CNN × 3 | 43.3 | 63.1 | 47.4 | 27.7 | 47.5 | 57.1 |
| | ATSS × 3 | 43.9 | 62.1 | 47.9 | 29.5 | 48.9 | 55.9 |
| | PAA × 3 | 44.1 | 61.7 | 47.9 | 28.6 | 48.6 | 58.9 |
| | RS R-CNN × 3 with Ref. NMS | 44.2 | 63.0 | 48.5 | 28.2 | 48.4 | 58.5 |
| | ATSS × 3 with Ref. NMS | 44.3 | 62.1 | 48.3 | 29.7 | 49.3 | 56.3 |
| | PAA × 3 with Ref. NMS | 44.6 | 61.6 | 48.7 | 28.9 | 49.1 | 59.5 |
| | RS R-CNN × 5 | 43.4 | 63.0 | 47.7 | 28.0 | 47.5 | 57.0 |
| | ATSS × 5 | 44.1 | 62.3 | 48.4 | 29.4 | 49.0 | 56.3 |
| | PAA × 5 | 44.4 | 62.0 | 48.4 | 28.9 | 49.0 | 59.2 |
| | RS R-CNN × 5 with Ref. NMS | 44.5 | 63.0 | 49.1 | 28.8 | 48.6 | 58.4 |
| | ATSS × 5 with Ref. NMS | 44.6 | 62.2 | 48.9 | 29.8 | 49.5 | 56.7 |
| | PAA × 5 with Ref. NMS | 45.0 | 61.8 | 49.3 | 29.4 | 49.6 | **59.9** |
| Mixtures of Experts | Vanilla MoE (RS R-CNN, ATSS, PAA) | 43.4 | 62.5 | 47.1 | 27.3 | 47.3 | 58.0 |
| | Vanilla MoE with Ref. NMS (RS R-CNN, ATSS, PAA) | 44.4 | 62.6 | 48.2 | 27.8 | 48.4 | 59.3 |
| | MoCaE (ATSS and PAA) - Ours | 44.8 | 62.4 | 49.2 | 29.4 | 49.1 | 57.6 |
| | MoCaE (RS R-CNN, ATSS, PAA) - Ours | **45.5** | **63.2** | **50.0** | **29.7** | **49.7** | 59.3 |
| | | +2.3 | +1.1 | +2.9 | +1.9 | +2.2 | +1.7 |

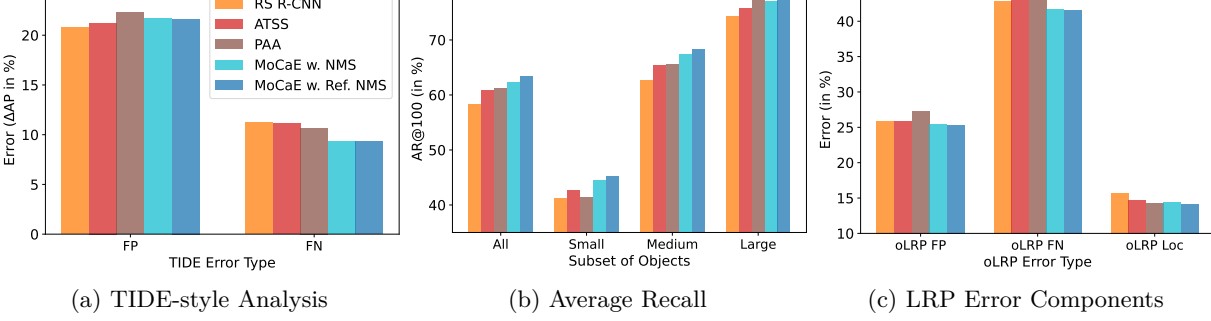

(a) TIDE-style Analysis      (b) Average Recall      (c) LRP Error Components

Figure A.7: The contribution of MoCaE on performance aspects

## D.4 Why does MoCaE Improve Performance of Single Detectors?

Here, we investigate how MoCaE improves the performance of single detectors and note two main takeaways: First, MoCaE combines the detectors in a way that the resulting MoE benefits from the complementary nature of the detectors; and as expected, second MoCaE mainly improves the recall of the single detectors.

### D.4.1 Complementary Nature of MoCaE

Ideally, an MoE aims to combine the individual components in such a way that they complement each other in different subsets of the input space to achieve a better performance. Here, we show that this is, in fact, the case also for MoCaE. Similar to our previous analyses, we combine ATSS, PAA and RS R-CNN and report the results on COCO *mini-test*. Differently, in order to be able to discuss the results, we utilize 12 supercategories such as *animal*, *food*, which are already present in COCO dataset. Fig. A.8 shows the results in which we can easily observe that different experts prevail on different supercategories. To illustrate, RS R-CNN (Oksuz et al., 2021a) is the best single model in the *appliance* supercategory, ATSS (Zhang et al., 2020b) is the best single model in the *kitchen* supercategory and PAA (Kim & Lee, 2020) is the best in the *indoor* supercategory. A noteworthy point is that MoCaE performs better than *any* of its components in *all of the supercategories*, achieving notable improvements over the entirety of the input space. This effectively

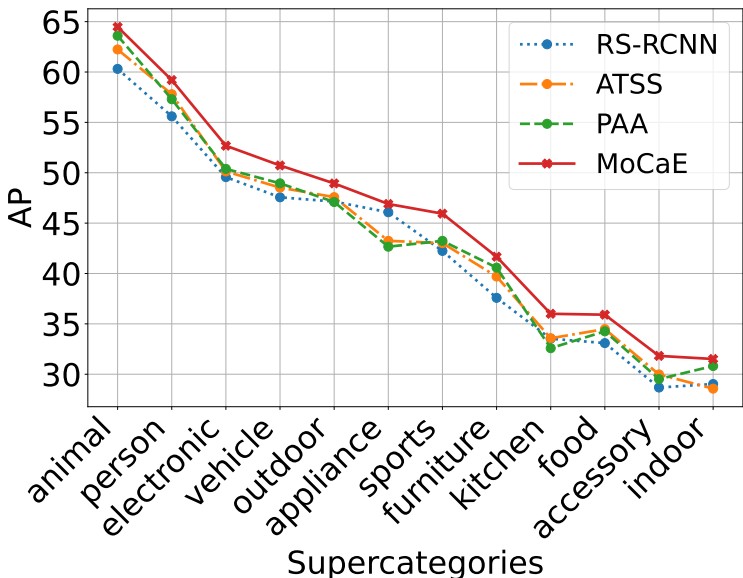

Figure A.8: AP scores of single models (RS-RCNN, ATSS, PAA) and MoCaE (consisting of RS-RCNN, ATSS, PAA) on different supercategories of the COCO *mini-test* dataset. MoCaE performs consistently higher than any of the single models on every supercategory.

shows the capability of MoCaE to leverage the complementary expertise of the single models to form a more accurate mixture.

### D.4.2 The Contribution of MoCaE on Different Performance Aspects

We now analyse what performance aspects, among localisation, recall and precision, are affected by MoCaE by expliting three different analyses tools from the detection literature. First, we use TIDE (Bolya et al., 2020) that defines oracle APs as the APs obtained when FP and false-negative (FN) errors are completely mitigated. Then, the difference between the oracle and actual AP correspond to the error of a detector for a specific performance aspect. More specifically, while obtaining Oracle FP AP, the FP detections are simply removed from the final detection set. As for Oracle FN AP, the FN objects are removed from the dataset. Fig. A.7(a) shows that MoCaE variants perform similar to the single detectors in terms of FP Error while they clearly outperform them on FN Error. This indicates that one of the contributions of MoE is to find the objects that are not detected by at least one of the individual detector as illustrated on Fig. 2 and Tab. 2. However, TIDE analysis does not provide insight on the localisation quality which is mainly targeted by Refining NMS. Therefore, in our second analysis we exploit AR defined as the average of the recall values over 10 IoUs from 0.50 to 0.95. Aligned with the observation in TIDE analysis, Fig. A.7(b) presents that MoCaE improves AR of the single detectors. Furthermore, the performance gain mainly originates from the improvement in small and medium objects as MoCaE does not improve the performance on large objects notably. This indicates that the resulting MoE is especially stronger than single detectors in more challenging object categories. Using Refining NMS further boosts the AR performance as it improves the localisation performance, which is critical for recalls with higher IoUs. To investigate the benefit of MoE in practical use-cases, we finally conduct an LRP analysis (Oksuz et al., 2021c; 2018) in Fig. A.7(c). In this figure, oLRP FP, FN and Loc correspond to the components Optimal LRP Error defined as 1-Precision, 1-Recall and the average IoU Error of TP detections. Aligned with our previous analysis, MoE mainly decreases the recall error and MoCaE with Refining NMS mainly contributes to the oLRP Loc component, outperforming all individual models in the end. These three different analyses confirm that MoCaE with NMS mainly decreases the recall error of the detector and using Refining NMS contributes to the localisation error.

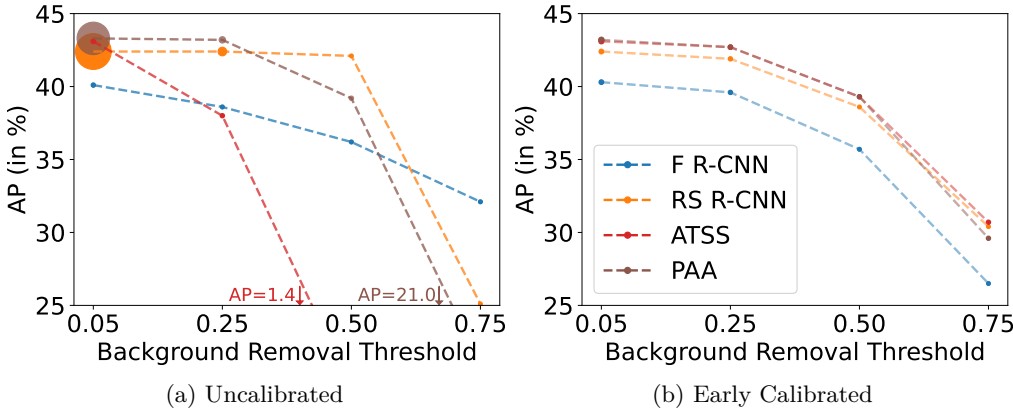

(a) Uncalibrated           (b) Early Calibrated

Figure A.9: The effect of background removal threshold on AP and NMS processing time for **(a)** uncalibrated and **(b)** early calibrated detectors on COCO. The area of the dots are proportional to the NMS processing time of the detectors. This threshold is typically set to 0.05, in which case PAA and RS R-CNN have large NMS processing time in **(a)**. Once uncalibrated, the detectors follow different trends and are sensitive to the threshold in terms of AP and NMS processing time. In **(b)**, early calibration (i) aligns the detector by reducing this sensitivity, and (ii) allows using 0.05 by both maximizing the AP and reducing NMS processing time for the over-confident PAA and RS R-CNN. F R-CNN refers to Faster R-CNN.

Table A.26: The values used in Fig. A.9. Early calibration regularizes the behaviour of the detectors by reducing their sensitivity to background removal threshold with respect to AP and NMS time. NMS time is measured in terms of ms using a single Nvidia 1080ti GPU.

| Background Removal Threshold | Detector | Uncalibrated | | Early Calibrated | |
|---|---|---|---|---|---|
| | | AP | NMS time | AP | NMS time |
| 0.05 | Faster R-CNN | 40.1 | 0.5 | 40.3 | 0.6 |
| | RS-RCNN | 42.4 | 35.4 | 42.4 | 0.6 |
| | ATSS | 43.1 | 0.6 | 43.1 | 0.7 |
| | PAA | 43.3 | 29.2 | 43.2 | 0.8 |
| 0.25 | Faster R-CNN | 38.6 | 0.4 | 39.6 | 0.5 |
| | RS-RCNN | 42.4 | 2.0 | 41.9 | 0.5 |
| | ATSS | 38.0 | 0.5 | 42.7 | 0.6 |
| | PAA | 43.2 | 1.1 | 42.7 | 0.5 |
| 0.50 | Faster R-CNN | 36.2 | 0.4 | 35.7 | 0.4 |
| | RS-RCNN | 42.1 | 0.4 | 38.6 | 0.4 |
| | ATSS | 19.3 | 0.4 | 39.3 | 0.5 |
| | PAA | 39.2 | 0.5 | 39.3 | 0.5 |
| 0.75 | Faster R-CNN | 32.1 | 0.4 | 26.5 | 0.4 |
| | RS-RCNN | 25.1 | 0.4 | 30.4 | 0.4 |
| | ATSS | 1.4 | 0.0 | 30.7 | 0.5 |
| | PAA | 21.0 | 0.4 | 29.6 | 0.5 |

### D.5 A Use Case for Early Calibration: Reducing the Sensitivity to Background Removal Threshold

Here, we investigate an additional use-case of early calibration in which it reduces the sensitivity of the detectors to background removal threshold in terms of both AP and efficiency. As AP provably benefits from more detections (Oksuz et al., 2023), detectors prefer a small background removal threshold as the first step of post-processing (Fig. 4(b)). *To illustrate,* 0.05 *is the common choice for COCO (Zhang et al., 2020b; Kim & Lee, 2020; Ren et al., 2017) and it is as low as* $10^{-4}$ *for LVIS (Chen et al., 2019; Gupta et al., 2019).* While this convention is preferred by AP, it can easily increase NMS processing time especially for over-confident detectors. This is because, for such detectors, the background removal step accepts redundant true-negatives (TNs), which should have been rejected. Hence, due to this large number of redundant TNs propagated to the NMS, NMS processing time significantly increases. To illustrate on PAA, which uses a threshold of 0.05

Table A.27: Experiments with Self-aware Object Detectors. We use the General Object Detection setting in (Oksuz et al., 2023). Please refer to the text for the details of the performance measures.

| Model Type | Self-aware Detector | DAQ ↑ | OOD Detection Performance BA ↑ | Acc. and Cal. for In-distribution | | | Acc. and Cal. for Domain-shift | | |
| | | | | IDQ ↑ | LaECE ↓ | LRP ↓ | IDQ ↑ | LaECE ↓ | LRP ↓ |
|---|---|---|---|---|---|---|---|---|---|
| Single Models | SA-RS-RCNN | 40.9 | **89.0** | 39.6 | 18.0 | 73.9 | 27.1 | 19.2 | 83.7 |
| | SA-ATSS | 40.9 | 87.9 | 39.6 | 17.9 | 73.9 | 27.3 | 20.6 | 83.5 |
| | SA-PAA | 40.6 | 87.3 | 39.0 | 17.8 | 74.4 | 27.1 | 21.0 | 83.6 |
| MoEs | Vanilla MoE | 42.5 | 88.7 | 39.9 | 18.1 | 73.6 | 29.2 | 19.6 | 82.1 |
| | MoCaE | **42.9** | 87.6 | **40.1** | **17.1** | **73.5** | **29.8** | **19.1** | **81.8** |

for COCO, NMS takes 29.2 ms/image on a Nvidia 1080Ti GPU, while it only takes $\sim 0.6$ ms/image for ATSS and Faster R-CNN (F R-CNN). This difference among the detectors can easily be noticed by comparing the areas of the dots at 0.05 in Fig. A.9(a)[*]. Specifically, we observed for PAA that $\sim 45K$ detections are propagated to the NMS per image on average. After early calibration, this number of detections from the same threshold reduces to $\sim 2K$ per image, which now enables NMS to take only 0.8 ms/image as ideally expected. Fig. A.9(b) presents that NMS takes consistently between 0.6 to 0.8 ms/image for all detectors as the behaviour of the detectors are aligned.

## D.6 Further Discussion on SAOD Task

**Details of the SAOD task** Self-aware object detection task (SAOD) (Oksuz et al., 2023) provides a comprehensive framework to evaluate the robustness of object detectors. Specifically, the performance aspects that are jointly considered in this task are out of distribution detection, calibration, domain shift as well as the accuracy. Specifically, SAOD evaluates an object detector in terms of the following criteria:

- Rejecting OOD images utilizing reliable image-level uncertainty estimates
- Yielding accurate and calibrated detections
- Being robust to domain shift under varying severities.

The evaluation is conducted using more than $150K$ single images, which is a large-scale test set enabling thorough evaluation. Specifically, the SAOD task utilizes the following split of the datasets for evaluating a detector in terms of the aforementioned performance aspects:

Table A.28: Object discovery on the out-of-distribution set (SinObj110K-OOD) from SAOD (Oksuz et al., 2023). Our gains in green are highlighted with respect to the best single model (underlined).

| Model Type | Detector | AR |
|---|---|---|
| Single Models | RS R-CNN | 41.3 |
| | ATSS | 42.0 |
| | PAA | 39.7 |
| MoEs | Vanilla MoE | 42.7 |
| | MoCaE - Ours | **45.6** |
| | | +3.6 |

- $D_{ID}$ : The in-distribution dataset consisting of images containing the same set of foreground objects as the training set.
- $T(D_{ID})$ : Domain-shift in-distribution dataset obtained through applying transformations from (Hendrycks & Dietterich, 2019) with severities 1, 3 and 5 on the in-distribution set.
- $D_{OOD}$ : The out-of-distribution dataset that contains only the objects of different foreground classes than that of the $D_{ID}$.

On this dataset, the ideal behavior expected from a robust object detector for a given input $X$ is:

- If $X \in D_{ID}$, "accept" the input and provide accurate and calibrated detections, any rejection is penalized.
- If $X \in T(D_{ID})$, with severities 1 and 3, "accept" the input and provide accurate and calibrated detections, any rejection is penalized.

---

[*]Tab. A.26 shows the exact values used to obtain Fig. A.9

- If $X \in T(D_{ID})$, with severity 5, provide the choice to "accept" the input though no penalty for rejections as transformed images might have severe deformities with respect to their original versions.
- If $X \in D_{OOD}$, "reject" the input and refrain from providing any detections, any accept is be penalized.

In association with these datasets, the authors also propose the following evaluation measures:

- Balanced Accuracy (BA) measures the OOD performance as the harmonic mean of TPR and TNR in this binary classification problem.
- LaECE measures the calibration performance, as discussed in App A.
- In-Distribution Quality (IDQ) combines accuracy and calibration performance as the harmonic mean of 1-LRP (Oksuz et al., 2018) and 1-LaECE on in-distribution data $D_{ID}$. Analogously, for domain-shifted data, $T(D_{ID})$, $IDQ_T$ is used.
- Finally, Distribution-Awareness Quality (DAQ) as the main performance measure of the task, unifies these measures. Specifically, DAQ is a higher the better measure, defined as the harmonic mean of BA, IDQ and $IDQ_T$.

**Implementation Details**   Based on the definition of this task, the authors also propose an algorithm to convert any detector to a self-aware one[*]. We follow the proposed algorithm to make RS R-CNN, ATSS and PAA, as well as Vanilla MoE and MoCaE self-aware. We use the General Object Detection setting in (Oksuz et al., 2023) as our models are trained on COCO, aligned with this dataset. We utilise the official SAOD code throughout our experiments by keeping all the settings.

**Discussion of the Results**   The results are presented in Tab. A.27 in which MoCaE improves DAQ measure, the main performance measures of this task, up to +2 points compared to the best single model while also showing notable improvements in terms of the rest of the measures. Furthermore, MoCaE also outperforms Vanilla MoE. This highlights that the MoCaE is more reliable than its counterparts in terms of the reliablity aspects considered within the SAOD (Oksuz et al., 2023) framework. When we examine the individual robustness aspects, we can easily see that MoCaE outperforms all of the single detectors and Vanilla MoE in terms of calibration and accuracy both on in-distribution and domain-shifted data. On the other hand, we observe that the OOD performance drops from $\sim 1$ BA compared to the best single model. In the following, we further discuss why this is the case and provide more insight.

**Is Lower OOD Performance a Pitfall of MoCaE or its Strength?**   MoCaE combines individual detectors to ideally benefit from the strength of each detector in the mixture. This commonly manifests itself to increase the recall performance of the individual components, i.e. Average Recall (AR), as we presented in Tab. 2 while motivating MoCaE as well as in Fig. A.7. AR consistently increases in such cases due to the fact that the detections of the individual experts are combined properly in in-distribution images.

Now, consider an alternative case, in which the input image is out-of-distribution (OOD), on which each detection is a FP as an OOD image does not contain an in-distribution object (as defined by (Oksuz et al., 2023)). In this specific case, combining the detectors properly might result in more FPs easily. To illustrate on a toy example, assume there are two images $X_1$ and $X_2$ and $k$ represents the number of maximum detections in an image, which is typically $k = 100$. Also assume that, an overconfident detector yields a high-confident FP on an out-of-distribution (OOD) image $X_1$ with $k - 1$ lower confident detections. As for $X_2$ this detector does not have a high-confident detection. This detector accepts the OOD image $X_1$ and rejects $X_2$. A second but an underconfident detector yields very low confidence detections on $X_1$ and one relatively confident detection on $X_2$. In contrary to $X_1$, this detector accepts the OOD image $X_2$ and rejects $X_1$. When we combine these two detectors without calibration, the resulting Vanilla MoE will mimic the overconfident detector, potentially rejecting $X_2$ and accepting $X_1$. Unfortunately, calibrating these two detectors in the form of MoCaE will have a confident detection in each image, potentially resulting in the acceptance of both OOD images. This is why Vanilla MoE mimics the most confident detector in Tab. A.27. In this specific case, the most confident detector, RS R-CNN has the largest BA, and consequently Vanilla MoE outperforms MoCaE in terms of OOD performance.

---

[*]We refer the reader to the Algorithm A.1 and Algorithm A.2 in (Oksuz et al., 2023) for the details.

Table A.29: Oracle MoE and pitfalls of MoCaE. Object detection performance on COCO *mini-test*. When the single methods have significant performance gap, then MoCaE might not perform well as the calibrators are imperfect. Oracle MoE is an MoE in which each expert is perfectly calibrated. As a result, all experts have 0 LaECE in Oracle MoE. Following Theorem 1, Oracle MoE outperforms all single detectors.

| Model Type | Method | AP | AP$_{50}$ | AP$_{75}$ | AP$_S$ | AP$_M$ | AP$_L$ | Unc. LaECE | Cal. LaECE |
|---|---|---|---|---|---|---|---|---|---|
| Single Models | RS R-CNN | 42.4 | 62.1 | 46.2 | 26.8 | 46.3 | 56.9 | 36.45 | 3.19 |
| | ATSS | 43.1 | 61.5 | 47.1 | 27.8 | 47.5 | 54.2 | 5.01 | 4.54 |
| | PAA | 43.2 | 60.8 | 47.1 | 27.0 | 47.0 | 57.6 | 11.23 | 1.09 |
| | YOLOv7 | 55.6 | 73.1 | 60.6 | 41.2 | 60.4 | 69.5 | 9.23 | 4.60 |
| | QueryInst | 55.9 | 75.4 | 61.3 | 38.5 | 60.8 | 73.2 | 4.54 | 3.19 |
| | DyHead | 56.8 | 75.6 | 62.2 | 42.8 | 60.6 | 71.0 | 10.04 | 6.26 |
| | EVA | 64.5 | 82.3 | 71.0 | 50.6 | 68.9 | 78.1 | 13.39 | 7.13 |
| | Co-DETR | 64.5 | 81.7 | 70.8 | 51.0 | 68.6 | 79.3 | 5.81 | 4.82 |
| MoEs of EVA and Co-DETR | Vanilla MoE | 64.6 | 82.3 | 71.3 | 50.7 | 68.8 | 79.0 | N/A | N/A |
| | MoCaE | 65.0 | 82.6 | 71.5 | 51.0 | 69.0 | 79.6 | N/A | N/A |
| | Oracle MoE | **81.9** | **96.6** | **91.1** | **72.7** | **85.2** | **92.1** | N/A | N/A |
| MoEs of RS R-CNN, ATSS, PAA EVA, Co-DETR | Vanilla MoE | 60.0 | 76.3 | 65.8 | 43.8 | 64.9 | 76.5 | N/A | N/A |
| | MoCaE | 61.9 | 78.8 | 69.0 | 49.0 | 66.6 | 76.4 | N/A | N/A |
| | Oracle MoE | **85.3** | **97.2** | **93.2** | **76.3** | **88.5** | **94.5** | N/A | N/A |
| MoEs of YOLOv7, QueryInst, DyHead EVA, Co-DETR | Vanilla MoE | 64.1 | 81.4 | 70.6 | 50.2 | 68.5 | 78.4 | N/A | N/A |
| | MoCaE | 63.1 | 80.1 | 69.5 | 49.9 | 68.0 | 77.5 | N/A | N/A |
| | Oracle MoE | **86.1** | **97.4** | **93.8** | **77.4** | **89.1** | **95.0** | N/A | N/A |
| MoEs of All Single Models | Vanilla MoE | 60.1 | 76.3 | 66.0 | 44.1 | 65.1 | 76.6 | N/A | N/A |
| | MoCaE | 61.7 | 78.3 | 68.1 | 48.7 | 66.9 | 76.3 | N/A | N/A |
| | Oracle MoE | **86.7** | **97.4** | **94.1** | **78.0** | **89.8** | **95.5** | N/A | N/A |

Following from the comparison of OOD performances of MoEs, the question arises why detectors generate high confidence detections on OOD images and whether there is a benefit of having them. More specifically, we conjecture that this is mainly a result of object discovery as the $D_{OOD}$ merely contains the classes that are not exactly present in $D_{ID}$, in which objects of semantically similar classes can still exist across the two. To support this claim, we design an experiment on the same OOD split. In this experiment, we use the bounding box annotations already present in the OOD split and check whether which of the models find the most number of objects. We do not distinguish among the classes as all the classes in OOD split (SinObj110K-OOD) pertains to the unknown class for all of the models we use. We report AR in Tab. A.28 to evaluate the models with respect to their object discovery characteristics. The results show that MoCaE outperforms all individual models and Vanilla MoE with a significant margin in terms of AR; demonstrating that it finds more objects compared to the other models. This is because, unsurprisingly, the confident detections of individual models correspond to the objects and combining them after calibration in MoCaE results in finding more unknown objects. As a result, while MoCaE does not perform the best in terms of image-level OOD detection, it is, in fact, a better alternative for object discovery.

## D.7 The Effect of Theorem 1 using an Oracle MoE

We proved in Theorem 1 that the calibration target that we designed in Eq. 4 provides the optimal AP under certain assumptions on post-processing. Here, we show that the assumptions are, in fact, not very strict and significantly higher AP values can easily be observed once the individual detectors in the mixture are perfectly calibrated as suggested in Theorem 1. To do so, we design an Oracle MoE, in which we replace the confidence score of the detections by the corresponding calibration target in Eq. 4 such that the detectors are now perfectly calibrated. In Oracle MoE, we use the standard NMS to keep its design simple. Tab. A.29 presents the results of this Oracle MoE combining different object detectors on COCO *mini-test*. When only two detectors, EVA and Co-DETR, are combined using Oracle MoE, we observe that the gain of the Oracle MoE compared to the best single model is more than 15AP, which is very significant. Furthermore, as the number of components increases, the performance of Oracle MoE consistently increases up to 86.7 AP and 97.4 AP$_{50}$ when all of the eight detectors (with very different performances) are combined. The

Table A.30: Detailed results on LVIS val set. The detectors have different characteristics in terms of exploiting Repeat Factor Sampling (RFS), backbone, the number of training epochs and the loss function. While Vanilla MoE does not yield gain, MoCaE enables a stronger MoE than all single detectors.

| Method | RFS | Backbone | Epoch | Instance Segmentation Performance | | | | | | Object Detection Performance | | | | | |
|---|---|---|---|---|---|---|---|---|---|---|---|---|---|---|---|
| | | | | AP | AP$_{50}$ | AP$_{75}$ | AP$_r$ | AP$_c$ | AP$_f$ | AP | AP$_{50}$ | AP$_{75}$ | AP$_r$ | AP$_c$ | AP$_f$ |
| Seesaw Mask R-CNN (Wang et al., 2020) | ✗ | ResNet-50 | 24 | 25.4 | 39.5 | 26.9 | 15.8 | 24.7 | 30.4 | 25.6 | 41.6 | 26.6 | 14.0 | 24.0 | 32.3 |
| RS Mask R-CNN (Oksuz et al., 2021a) | ✓ | ResNet-50 | 12 | 25.1 | 38.2 | 26.8 | 16.5 | 24.3 | 29.9 | 25.8 | 39.7 | 27.8 | 15.1 | 24.5 | 32.0 |
| Mask R-CNN (He et al., 2017) | ✓ | ResNeXt-101 | 12 | 25.4 | 39.2 | 27.3 | 15.7 | 24.7 | 30.4 | 26.6 | 42.1 | 28.5 | 15.4 | 25.2 | 33.1 |
| Vanilla MoE | N/A | N/A | N/A | 25.2 | 38.3 | 26.8 | 16.5 | 24.3 | 29.9 | 25.9 | 39.8 | 27.9 | 15.1 | 24.5 | 32.2 |
| | | | | −0.2 | −1.2 | −0.5 | 0.0 | −0.4 | −0.5 | −0.7 | −2.3 | −0.6 | −0.3 | −0.7 | −0.9 |
| MoCaE (Ours) | N/A | N/A | N/A | **27.7** | **42.8** | **29.4** | **18.2** | **27.3** | **32.4** | **29.1** | **44.8** | **31.4** | **17.0** | **27.9** | **35.8** |
| | | | | +2.3 | +3.3 | +2.1 | +1.7 | +2.4 | +2.0 | +2.5 | +1.6 | +2.9 | +1.6 | +2.7 | +2.7 |

significance of these AP values validates Theorem 1 from the practical perspective. On the other hand, once the performance gap among the combined detectors increases, we observe that Vanilla MoE and MoCaE do not perform well, which we discuss in the following section.

## D.8 Limitations of MoCaE

**On the importance of performance difference among the detectors**   As discussed in Sec. 4.4 in the paper, the main limitation of MoCaE (as well as Vanilla MoE) is that it benefits from combining similarly performing detectors. This is presented in Tab. A.29 in which we can see that combining low-performing detectors (e.g., RS R-CNN or YOLOv7) with EVA and Co-DETR decreases the performance of MoE to an AP less than the best single detector. On the other hand, as just presented in App. D.7, this is not the case for Oracle MoE. This suggests that while the calibration decreases the LaECE of the single detectors significantly (Tab. A.29), they are still far from being well-calibrated to construct an accurate MoE. This is because, we keep the capacity of the calibrators low, that is Class-agnostic LR and IR are monotonically increasing function of the predicted confidence; thereby preserving the ranking among the detections and the AP of each detector. On the other hand, this capacity limitation might not result in better MoEs than the individual components once their performance gap is high. Furthermore, the Oracle MoE consistently improves upon the individual detectors even the performance gap is significantly high as shown in Tab. A.29. This demonstrates the importance of the calibration quality in obtaining an accurate MoE as well as the potential to improve the object detectors using MoEs, which is an open problem.

**MoCaE requires more resource than single models**   Another limitation of MoCaE stems from using multiple models during inference, which is also the case for DEs. Still, as each detector can process the input in parallel, the overhead introduced by calibration would be negligible when a separate GPU is allocated for each detector. As an example, on an Intel i7 3.60GHz CPU, obtaining the isotonic regression calibrator and inference take 12.0 ms and 0.30 ms/image respectively. This inference cost is negligible as it increases the inference time of ATSS only by 0.2% and EVA by 0.005%. Besides, comparing Fig. 1(b) with Fig. 1(a), we showed that calibration balances the contribution of the detectors to MoE. However, PAA, as the most accurate detector. has still the lowest contribution with %25.61 in Fig. 1(b). This is because LaECE after calibration in Tab. 1 is still non-zero for the detectors. Therefore, better calibration methods could give rise to more accurate MoEs.

## D.9 Full Version of the Tables Pruned in the Paper

Tab. A.30 presents the detailed results on LVIS dataset for instance segmentation; Tab. A.31 includes the performance of all classes for DOTA dataset; Tab. A.32 shows the performance over different object scales; and Tab. A.33 includes COCO minitest results for common object detectors as well as their main differences in terms of pretraining data and the backbone. These tables are excluded from the main paper due to the space limitation.

Table A.31: The performance of all classes on DOTA v1.0. $AP_{50}$ is the main performance measure of DOTA.

| Detector | All | Plane | Baseba. | Bridge | Ground-t. | Small-veh. | Large-veh. | Ship | Tennis | Basket. | Storag. | Soccer | Roundab. | Harbor | Swimm. | Helico. |
|---|---|---|---|---|---|---|---|---|---|---|---|---|---|---|---|---|
| RTMDet | 81.32 | 88.04 | 86.20 | 58.50 | 82.43 | 81.21 | 84.87 | 88.70 | 90.89 | 88.75 | 87.33 | 72.12 | 70.85 | 81.16 | 81.49 | 77.24 |
| LSKN (prev. SOTA) | 81.85 | 89.69 | 85.70 | 61.47 | 83.23 | 81.37 | 86.05 | 88.64 | 90.88 | 88.49 | 87.40 | 71.67 | 71.35 | 79.19 | 81.77 | 80.85 |
| Vanilla MoE | 80.60 | 87.76 | 83.27 | 61.77 | 78.25 | 81.26 | 85.33 | 88..34 | 89.93 | 85.54 | 86.35 | 70.98 | 65.92 | 84.28 | 82.45 | 77.57 |
|  | −1.25 | −1.93 | −2.93 | +0.30 | −4.98 | −0.11 | −0.72 | −0.36 | −0.11 | −0.96 | −1.05 | −1.14 | −5.43 | +3.12 | +0.68 | −3.28 |
| MoCaE (Ours) | 82.62 | 89.09 | 86.47 | 61.38 | 83.28 | 81.43 | 85.03 | 88.72 | 90.86 | 88.31 | 87.11 | 75.50 | 74.12 | 84.49 | 81.63 | 81.93 |
|  | +0.77 | −0.60 | +0.27 | −0.09 | +0.05 | +0.06 | −1.02 | +0.02 | −0.03 | −0.44 | −0.29 | +3.38 | +2.77 | +3.33 | −0.14 | +1.08 |

Table A.32: Effect of calibration on MoE performance. All MoEs use Late calibration with standard NMS. While combining uncalibrated detectors do not provide notable gain over the single detectors, calibration is essential for a strong MoE resulting in up to $\sim 1.5$ AP gain over single detectors.

| Model Type | Calibration | Combined Detectors | | | Object Detection Performance | | | | | |
|---|---|---|---|---|---|---|---|---|---|---|
|  |  | RS R-CNN | ATSS | PAA | AP | $AP_{50}$ | $AP_{75}$ | $AP_S$ | $AP_M$ | $AP_L$ |
| Single Models | N/A | ✓ |  |  | 42.4 | 62.1 | 46.2 | 26.8 | 46.3 | 56.9 |
|  | N/A |  | ✓ |  | 43.1 | 61.5 | 47.1 | 27.8 | 47.5 | 54.2 |
|  | N/A |  |  | ✓ | 43.2 | 60.8 | 47.1 | 27.0 | 47.0 | 57.6 |
| Mixtures of Experts | ✗ | ✓ | ✓ |  | 42.4 | 62.1 | 46.3 | 26.8 | 46.3 | 56.9 |
|  | ✓ | ✓ | ✓ |  | 44.1 | 63.0 | 48.4 | 28.5 | 48.4 | 56.8 |
|  | ✗ | ✓ |  | ✓ | 43.4 | 62.5 | 47.1 | 27.3 | 47.3 | 58.0 |
|  | ✓ | ✓ |  | ✓ | 44.0 | 62.7 | 47.9 | 28.2 | 48.2 | 58.1 |
|  | ✗ |  | ✓ | ✓ | 43.3 | 60.9 | 47.2 | 27.1 | 47.2 | 57.6 |
|  | ✓ |  | ✓ | ✓ | 44.4 | 62.5 | 48.5 | 29.2 | 48.5 | 57.3 |
|  | ✗ | ✓ | ✓ | ✓ | 43.4 | 62.5 | 47.1 | 27.3 | 47.3 | 58.0 |
|  | ✓ | ✓ | ✓ | ✓ | 44.7 | 63.1 | 48.9 | 29.2 | 49.0 | 58.2 |

## D.10 Reliability Diagrams of the Utilized Models

For further insights, we include the reliability diagrams of the object detectors and instance segmentation models in Fig. A.10-Fig. A.14. The figures show that the calibration performance of the detectors improves after using post-hoc calibrators, enabling us to construct an effective MoE using MoCaE. Here, we use 500 images to calibrate the detectors. This is the reason of the small difference between LaECE values of Fig. A.10 and Tab. 1 which uses all 2500 images in COCO minival for calibration.

Table A.33: Object detection performance on COCO *test-dev* and *mini-test* using strong object detectors. The gains are reported compared to the best single model as underlined. MoCaE maintains the significant AP boost also for this challenging setting as well.

| Method | Pretraining Data | Backbone | COCO test-dev | | | | | | COCO minitest | | | | | |
|---|---|---|---|---|---|---|---|---|---|---|---|---|---|---|
| | | | AP | AP$_{50}$ | AP$_{75}$ | AP$_S$ | AP$_M$ | AP$_L$ | AP | AP$_{50}$ | AP$_{75}$ | AP$_S$ | AP$_M$ | AP$_L$ |
| YOLOv7 (Wang et al., 2022a) | None | L-size conv. | 55.5 | 73.0 | 60.6 | 37.9 | 58.8 | 67.7 | 55.6 | 73.1 | 60.6 | 41.2 | 60.4 | 69.5 |
| QueryInst (Fang et al., 2021) | None | Swin-L | 55.7 | 75.7 | 61.4 | 36.2 | 58.4 | 70.9 | 55.9 | 75.4 | 61.3 | 38.5 | 60.8 | 73.2 |
| DyHead (Dai et al., 2021) | ImageNet22K | Swin-L | 56.6 | 75.5 | 61.8 | 39.4 | 59.8 | 68.7 | 56.8 | 75.6 | 62.2 | 42.8 | 60.6 | 71.0 |
| Vanilla MoE | N/A | N/A | 57.6 +1.0 | 76.6 +0.9 | 63.2 +1.4 | 40.0 +0.6 | 60.9 +1.1 | 70.8 −0.1 | 57.7 +0.9 | 76.3 +0.7 | 62.9 +0.7 | 42.6 −0.2 | 62.7 +1.9 | 72.8 −0.4 |
| MoCaE (Ours) | N/A | N/A | **59.0** +2.4 | **77.2** +1.5 | **64.7** +2.9 | **41.1** +1.7 | **62.6** +2.8 | **72.4** +1.5 | **58.9** +2.1 | **76.8** +1.1 | **64.3** +2.1 | **44.7** +1.9 | **63.6** +2.8 | **74.1** +1.1 |

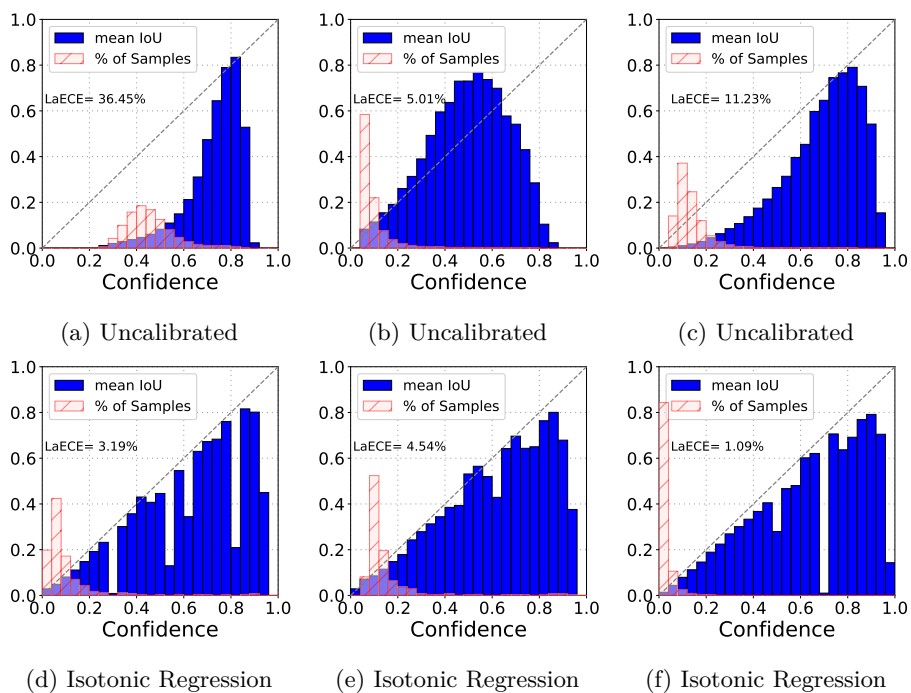

(a) Uncalibrated    (b) Uncalibrated    (c) Uncalibrated

(d) Isotonic Regression    (e) Isotonic Regression    (f) Isotonic Regression

Figure A.10: Reliability diagrams of the detectors presented on Tab. 3. **(a)** Uncalibrated RS R-CNN, **(b)** Uncalibrated ATSS, **(c)** Uncalibrated PAA, **(d)** RS R-CNN calibrated by Isotonic Regression, **(e)** ATSS calibrated by Isotonic Regression, and **(f)** PAA calibrated by Isotonic Regression.

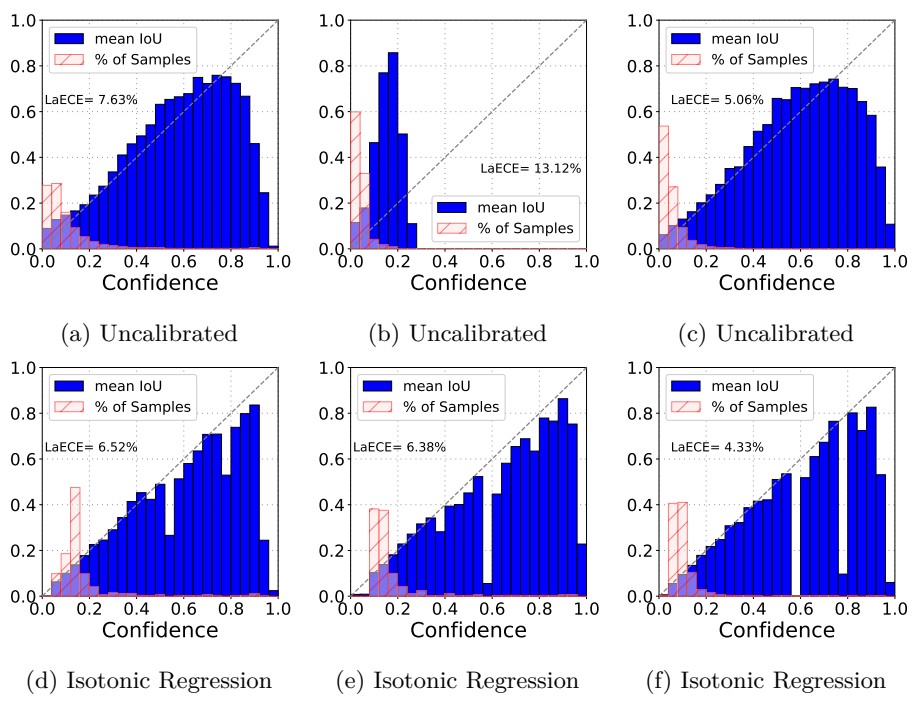

Figure A.11: Reliability diagrams of the detectors presented on Tab. 5. **(a)** Uncalibrated Co-DETR, **(b)** Uncalibrated BRS-DETR, **(c)** Uncalibrated DINO, **(d)** Co-DETR calibrated by Isotonic Regression, **(e)** BRS-DETR calibrated by Isotonic Regression, and **(f)** DINO calibrated by Isotonic Regression.

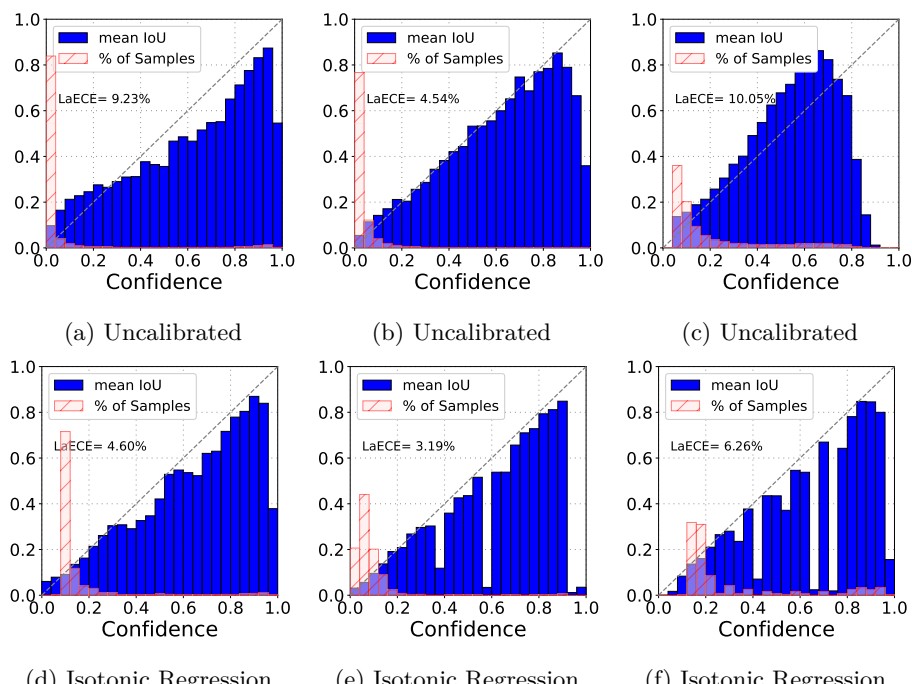

Figure A.12: Reliability diagrams of the detectors presented on Tab. 7. **(a)** Uncalibrated YOLOv7, **(b)** Uncalibrated QueryInst, **(c)** Uncalibrated DyHead, **(d)** YOLOv7 calibrated by Isotonic Regression, **(e)** QueryInst calibrated by Isotonic Regression, and **(f)** DyHead calibrated by Isotonic Regression.

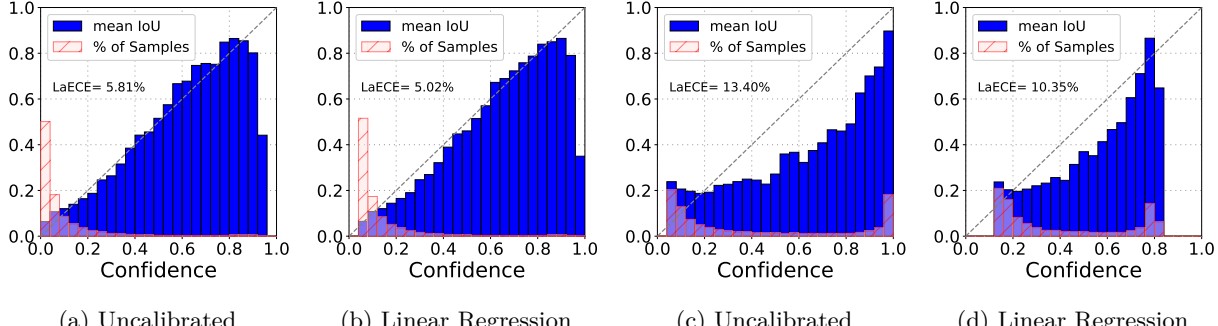

(a) Uncalibrated  (b) Linear Regression  (c) Uncalibrated  (d) Linear Regression

Figure A.13: Reliability diagrams of the detectors from Tab. 8. **(a)** Uncalibrated Co-DETR, **(b)** Co-DETR calibrated by Linear Regression, **(c)** Uncalibrated EVA, and **(d)** EVA calibrated by Linear Regression.

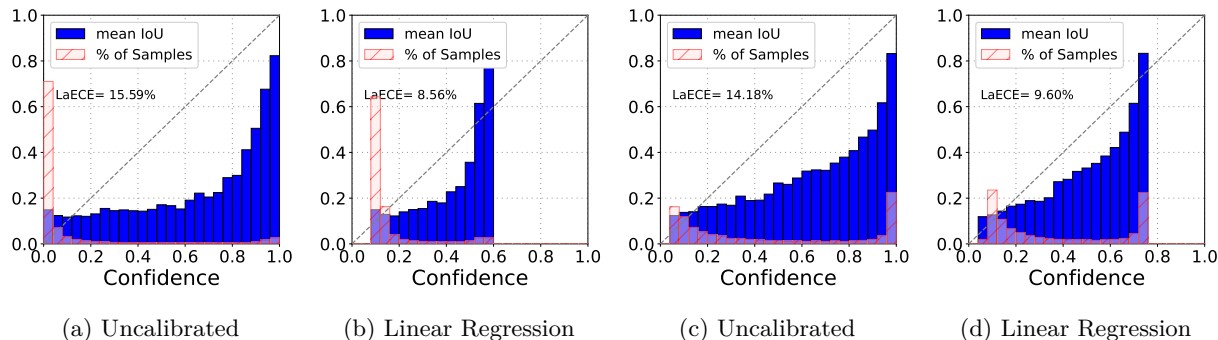

(a) Uncalibrated  (b) Linear Regression  (c) Uncalibrated  (d) Linear Regression

Figure A.14: Reliability diagrams of the detectors from Tab. 12. **(a)** Uncalibrated Mask2Former, **(b)** Mask2Former calibrated by Linear Regression, **(c)** Uncalibrated ViTDet, and **(d)** ViTDet calibrated by Linear Regression.

