# OpenReview forum: "MoCaE: Mixture of Calibrated Experts Significantly Improves Object Detection"
_TMLR — Accepted by TMLR_

### Review · Reviewer_yiuu · 2024-08-11

**Summary Of Contributions:**

1. This paper reveals that the simple model ensemble of different detectors may hurt the final performance due to the vastly different confidences of these detectors. The diversity in training regimes for these detectors leads to such prediction discrepancy.

2. To address this issue, this paper proposes a MoE method called MoCaE. MoCaE first calibrates the results of each expert and fuse these calibrated predictions with an elaborate Refining NMS.

3. Extensive experimental results are conducted to demonstrate the effectiveness of the proposed framework.

**Audience:**

Yes

**Broader Impact Concerns:**

There are no ethical implications of the work.

**Claims And Evidence:**

Yes

**Requested Changes:**

Please see the Weaknesses listed above.

**Strengths And Weaknesses:**

Strengths:

1. The motivation is clear. The observation that confidence discrepancy leads to degraded performance of MoE is interesting.

2. The qualitative and quantitative results demonstrate advantages over prior baselines.

3. The paper is easy to read and understand.


Weaknesses:

1. Can you show more examples of various detectors (Faster-RCNN [1], Cascade-RCNN [2], GFL [3], Deformable-DETR [4], DINO [5], and Co-DETR [6]) just like Fig 3? Besides, can we just simply calibrate each expert by score normalization?

2. The post-hoc calibrator described in Section 3.2.1 seems an additional branch that predicts the IoU score for each object. Please compare your method with the IoU branch design (only the IoU branch is trainable and it receives the object features as inputs).

3. What's the performance upper bound of MoCaE (using GT IoU scores to calibrate experts)?

4. The novelty of Refining NMS is limited as it seems a combination of Soft-NMS and Score Voting.

5. This method employs a held-out validation set for recalibration and this can lead to data leakage. Therefore, the comparisons on the COCO minitest are not fair. The performance of models that utilize the COCO train for calibration should be presented.

[1] Faster r-cnn. ICCV 2015.

[2] Cascade r-cnn: Delving into high quality object detection. CVPR 2018.

[3] Generalized focal loss: Learning qualified and distributed bounding boxes for dense object detection. NeurIPS 2020.

[4] Deformable detr: Deformable transformers for end-to-end object detection. ICLR 2020.

[5] Dino: Detr with improved denoising anchor boxes for end-to-end object detection. ICLR 2023.

[6] Detrs with collaborative hybrid assignments training. ICCV 2023.

---

> ### Author Response · Authors · 2024-09-18
> **On more reliability diagrams, Post-hoc calibration and data leakage**
>
> 1. As requested, we plotted the reliability histogram of the detectors in the revised version. Please refer to Fig.A.10-14 and Section D.10 at the end of Appendix D. These figures include the reliability diagram before and after calibration for 13 different detection and segmentation methods and confirm that the calibration for each improves after using post-hoc calibrators. As for the **score normalization**, we assume that the reviewer suggests normalizing the scores to a predefined distribution, such as a gaussian with a mean of 0.50 and standard deviation of 0.15 also by ensuring the scores are between [0,1]. However, please note that such a score normalization-based approach would not consider the quality of the detectors. As an example, please consider combining two detectors: Detector A and Detector B. Assume that while Detector A can perfectly find the objects perfectly, Detector B only produces false positives. In that case, score normalization will keep some detections from Detector B, resulting in a lower AP than that of Detector A. However, as Theorem 1 in the paper proves, if the calibrator is perfect, MoCaE will select the detections from Detector A, avoiding the failure case of score normalization.
>
> 2. We thank the reviewer for his/her invaluable idea. Actually, we had tried this idea in the early stage of the project, however, we couldn’t make it work. Specifically, we used Faster R-CNN and saved the features of each region-of-interest in the last layer. Then, we trained a model to regress the IoU of the detections on the COCO minival set. At inference, we replaced the confidence score of the detection by the output of the model. We extensively searched for the hyperparameters including learning rate and weight decay, used different loss functions (including generalized focal loss to address the underlying imbalance of the IoUs) and investigated different architectures such as a linear layer or an MLP. However, **the performance of Faster R-CNN dropped by more than 10AP using the best performing setting of the hyperparameters**. We conjectured that this is because a small validation set does not contain enough variation to enable the model to learn the mapping from the feature space to the IoUs. Furthermore, it is unclear (and not investigated in the literature yet) how to include this additional branch for detectors like transformer-based detectors or rotated bounding detectors. To conclude, it is non-trivial to train such a model for 19 different detectors that we used for 5 different detection tasks in our paper.
>
> 3. We already discussed this in the paper. Please refer to Table A.29 and Appendix D.7. For example, while MoCaE of EVA and Co-DETR yields 65.0 AP with the post-hoc approaches, the MoCaE of the same detectors would have 81.9 AP with a perfect calibrator. This presents the importance of better calibration approaches for object detectors while combining them in the form of MoEs.
>
> 4. While we combine two existing approaches to obtain Refining NMS, we want to mention that this is not the core contribution of our paper. The main problem that we aim to solve in this paper is how to combine off-the-shelf detectors in the form of an MoE in a principled way such that the resulting MoE performs better than these individual detectors. For that purpose, we show that aggregating high quality detections from multiple experts using Refining NMS is a much better way compared to using standard NMS.
>
> 5. We kindly want to mention that, post-hoc detectors are trained on a held-out validation set in the literature [C,D]. This is mainly because obtaining post-hoc calibrators on the training set will yield biased calibrators as the deep learning models are often overconfident on the training set [C]. Therefore, similar to [D,E], we split COCO val set as minival and minitest ensuring that **there is no data leakage**. Furthermore, this is only one setting in our paper we used and we list the other settings as follows:
> - Tables 7 and 8 are obtained by submitting the detections on the COCO test-dev split to COCO evaluation server
> - Table 9 is obtained by submitting the detections on the DOTA test split to DOTA evaluation server
> - ODinW-35 results in Table 10 are obtained by submitting the detections on the ODinW-35 test split to evaluation server
> - Objects45K is a different dataset for domain shift in Table 13
>
> These results further demonstrate the effectiveness of our results on different datasets and tasks in addition to the COCO minitest dataset.
>
> [C] Guo et al., On Calibration of Modern Neural Networks, ICML 2017
> [D] Kuzucu et al., On Calibration of Object Detectors: Pitfalls, Evaluation and Baselines, ECCV 2024.
> [E] Kuppers et al., Parametric and multivariate uncertainty calibration for regression and object detection. Safe Artificial Intelligence for Automated Driving Workshop in The European Conference on Computer Vision, 2022.

---

> > ### Comment · Reviewer_yiuu · 2024-09-19
> >
> > Thank you for your response, it helped me to understand your work much better and resolved several concerns.
> >
> > I would like to clarify a couple of more questions if that's possible:
> >
> > 1. Will the post-hoc calibrators be biased if trained on a subset of the training images (e.g., randomly selected 2.5k images)?
> >
> > 2. Can MoCaE improve the performance of a multi-detector ensemble using multi-scale testing (TTA)? Specifically, how does the performance of MoCaE + detector A with TTA + detector B with TTA, as well as to detector A with TTA and detector B with TTA individually?

---

> > > ### Author Response · Authors · 2024-09-23
> > > **On post-hoc calibrators on training & multi-scale testing**
> > >
> > > We thank the reviewer for the response and thought-provoking questions. We submitted a new revision of the paper with the new results to address these questions and provide more insights on MoCaE in the final version of the paper. Below, we present a summary of the changes and our results:
> > >
> > > 1. In the new revision of the paper, we include experiments in which we obtain post-hoc calibrators on a subset of the training set and construct MoCaE using these calibrators. Specifically, we experiment this using COCO dataset in Table A.19 and using LVIS dataset in Table A.20. Our main observation is that *while MoCaE constructed on train and validation sets perform almost on-par on COCO, there is a significant performance gap ($1.0$ AP) in favor of using validation set for LVIS*. This is mainly because LVIS is a long-tailed dataset for which the models tend to overfit more as shown in the literature. We include more detailed discussion in the last paragraph of Section D.2.1 (pages 28,29). As a result, this validates the fact that ideally the calibrators should be obtained on a held-out validation set. Having said that, our results consistently show that using a subset of training set is more useful (e.g., when there is no held out validation set.) compared to using Vanilla MoE.
> > >
> > > 2. In the new revision of the paper, we clearly show that MoCaE is complementary to multi scale testing. Section D.2.3 on page 30 (and Table A.22) provides the results and the details. To summarise, MoCaE with multi scale testing outperforms (i) the single best model with multi scale testing by $\sim 2$ AP; (ii) MoCaE without multi scale testing by $\sim 1$AP; and (iii) Vanilla MoE with multi scale testing by $2$AP.

---

### Review · Reviewer_iRbf · 2024-08-13

**Summary Of Contributions:**

The paper introduces a novel approach to enhancing object detection performance by addressing the issue of miscalibration in the combination of multiple expert detectors. The authors propose the Mixture of Calibrated Experts (MoCaE) framework, which first calibrates the predictions of individual detectors and then refines them using a new strategy called Refining Non-Maximum Suppression (NMS). This method effectively balances the contributions of each expert, leading to significant improvements in detection accuracy. The paper's extensive experiments across multiple detection tasks demonstrate MoCaE's effectiveness, achieving state-of-the-art results on benchmarks such as COCO and DOTA. The work is supported by a theoretical analysis that explains why calibration is crucial for optimal performance in mixtures of experts.

**Audience:**

Yes

**Broader Impact Concerns:**

There are no significant ethical concerns directly arising from this work.

**Claims And Evidence:**

Yes

**Requested Changes:**

1. "predicted confidences do not match the accuracy." should be changed with in a more precise to illustrate "calibration".
2. add experiments for different experts as in weakness.

**Strengths And Weaknesses:**

## Strengths

**Robust Experimental Results**: The paper presents extensive testing across various tasks (e.g., COCO, DOTA), consistently showing improved performance and achieving state-of-the-art results.

**Clear and Readable Presentation**: The paper is well-organized and easy to follow, with clear explanations, figures, and tables that effectively convey the proposed approach and its results.

## Weakness

Here are two weaknesses of the paper:

**Limited Novelty**: The novelty of the proposed calibration technique is somewhat trivial, as similar calibration methods have been extensively studied in the context of classification. The approach might not offer significant new insights beyond existing work.

**Limited Experimental Scope**: The experiments involve only three distinct models with different calibration characteristics. The paper would benefit from additional experiments with varying combinations of models, such as all low-confidence models or all high-confidence models, to better evaluate the effectiveness of MoCaE compared to Deep Ensembles (DE).

---

> ### Author Response · Authors · 2024-09-18
> **On the Novelty and Experimental Scope**
>
> 1. **On the Limited Novelty**: We want to clarify that the main aim of our paper is not proposing a new calibration approach. *Our main contribution is that we show that it is non-trivial to combine off-the-shelf detectors belonging to different families in the form of an MoE due to their inherent differences. To address this and obtain an effective MoE, we use calibration just as a means.* Also, we believe that simplicity of the used calibration approach is a strength not a weakness as our perspective allows obtaining effective MoEs in an efficient and easy manner.
>
> 2- **On the Limited Experimental Scope**: In the revision of the paper, we added 4 new detection and segmentation methods including BRS DETR, DINO, VitDet and Mask2Former.  Among these experiments, we have detectors with similar confidence distributions in our paper. To show that we plotted the reliability diagrams of 13 different detectors in Fig.A.10-14 (Section D.10) at the end of appendix D. Specifically, as an example VitDet and Mask2Former in Fig. A.14 are both overconfident as the blue histograms lie below the diagonal in the same figure. We show in Table 12 in the revised paper that combining these two overconfident detectors using MoCaE improves the performance of the best model by 1.1 mask AP. Please also note that if the detectors with very similar confidence score distributions are combined, a calibrated MoE will behave similarly to Vanilla MoE as the calibration will not be needed in the first place. One extreme example is the deep ensembles in which multiple copies of the same model are trained from different initializations. We show in Table A.18 that as expected calibration does not have much effect in this particular case, but doesn’t harm the performance either. We also provide another example in the revised paper (please refer to the blue paragraph in Section 4.1) that calibration is not very critical once different detectors with very similar confidence distributions (the MoE of Co-DETR and DINO pair in Table 4) are combined. Again we can see in Table 4 that, the performance is not affected notably. Therefore, also considering the consistent positive effect of our Refining NMS, these experiments suggest that MoCaE can be used even when the confidence score distributions are similar.
>
> We also want to kindly mention that we experiment MoCaE on 5 different detection tasks (object detection, rotated object detection, open vocabulary object detection, instance segmentation, self-aware object detection) using 19 different detectors after including the recent detectors in the revised version. We believe that our experiments are very comprehensive now.
>
> 3- **"predicted confidences do not match the accuracy." should be changed**: We edited the specified text in the introduction and explicitly mentioned that the mismatch phenomenon corresponds to miscalibration. Please refer to the blue text in the first page of the revised paper.

---

### Review · Reviewer_1nWj · 2024-09-04

**Summary Of Contributions:**

The work propose simple approach to combine multiple detection experts for improved detection performance. Naively combining different experts results in inferior performance due to difference in confidences of predictions of different experts, as some of the expert can dominate the results with high confident predictions. The work proposes calibration of the confidence followed by refining non maximal suppression for better selection of boundary boxes. Overall it improves over different baselines achieving state of the art results.

**Audience:**

Yes

**Claims And Evidence:**

Yes

**Requested Changes:**

Writing changes - The work should improve discussions of LAECE and score voting as presented in weakness 2. The current version is not self contained and is difficult to understand from the main paper what is happening. The reader will have to jump to supplement to understand it better. In general the main paper should be self contained.

Clarification on using old detectors as the main result as in limitation 1 and 3 : Overall I liked the work, however the limitation that it performs best with old detectors makes it less attractive. Further results using Mask2Former in instance segmentation result would make the claim stronger (limitation 3). Further clarification on why the learning of calibration cannot happen during training will be helpful.

**Strengths And Weaknesses:**

The strengths include -
1. The work proposes simple method of calibrating individual detection experts confidence score for improved results in mixture of experts.

2. The work has extensive results with different datasets and models and usecases as well. For eg the work also shows result with open vocabulary detection and instance segmentation.

3. The paper is nicely written with good illustrations on challenges faced by naive mixture of experts and how the work tackles it (fig 1 and 2).

The weaknesses include -
1. Choice of detectors - The main result table (table 1) shows result with detectors which are old/ outdated (RS R-CNN- 2021, ATSS - 2020, PAA - 2020). While it shows promising results with these detector with the MoCAE, it would be good to have result on the main table with new detectors. The result with new detectors like co-DETR and EVA are presented quite late in table 6, also has less of improvement with proposed mixture of calibrated experts.

2. Discussion not self contained: The work discusses about Localisation-aware Expected Calibration error (LAECE) in equation 1, which is not self contained. For example what does the bins do, what is IoU^c (j) etc. The work points out to supplement for details, however for the discussion is complete. The texts should be self contained. Similarly score voting , eq 3, does not have proper discussion as what is happening and how the final bounding box after NMS is obtained. The work says refer to App. D, but again the discussion is not complete and self contained.

3. Instance segmentation results with quite old methods: Although the experiments work with mixture of experts of Mask R-CNN (2017) and its variant (2020, 2021), the results with SoTA works such as Mask2Former would boost the contribution claim.

4. Learning calibration via linear regression during training:  The post processing for Mixture of calibrated experts is applied on part of the val splits, and does not require training. However one could have the same objective in the training, using training data, while learning individual experts. Sure learning on part of val split with less data is fast and efficient, however one can also better learn the calibration regression model on train data (maybe with bigger models). Any reason why this is not preferred ?

---

> ### Author Response · Authors · 2024-09-18
> **Recent Detectors, Segmentation Methods & Self-Contained Content**
>
> 1. Following the reviewer’s suggestion, we combined three different recent transformer-based detectors in the revised version (Co-DETR, ICCV 2023;  DINO, ICLR 2023 and BRS-DETR, ECCV 2024). Specifically, we added Table 4 showing that their LaECE improves after calibration (as requested by the reviewer as Table 1) and Table 5 including the accuracy of the combination of each pair of these detectors as well as all three. We also extended our discussion on calibration in Section 4.1 by referring to these tables.
>
> 2. As recommended by the reviewer, we included more discussion on LaECE in Section 3.1 and similarly on Refining NMS in Section 3.2.2 in the revised version of the paper. As a result of providing detailed explanation on Refining NMS in Section 3.2.2, we also removed Appendix D.
>
> 3. As suggested we combined Mask2Former (CVPR 2022) with VitDet (ECCV 2022) in Table 12 of the revised paper. We also included a discussion on that Table in Section 4.2 of the revised paper (specifically the blue paragraph under instance segmentation). To summarize, our results show that MoCaE improves Mask2Former by 1.1 AP once it is combined with VitDet, also by outperforming Vanilla MoE; further demonstrating the effectiveness of our approach.
>
> 4. Conceptually, this can work. In fact, there are training-time approaches in the classification literature [A] with better calibration than post-hoc ones. However, as [B] showed this is not the case for detection yet and existing post-hoc approaches outperform training-time calibration approaches significantly. Also from a practical perspective, this will require retraining all the detectors in the mixture considering the calibration target. This is both impractical as it will require resources and it is challenging as the architectures of object detectors differ. On the other hand, using post-hoc calibrators abstracts away the underlying object detector architecture and does not require retraining them. That is why post-hoc calibrators are more preferable over training time approaches for object detection at the moment.
>
> [A] Mukhoti et al., Calibrating deep neural networks using focal loss, NeurIPS 2020.
> [B] Kuzucu et al., On Calibration of Object Detectors: Pitfalls, Evaluation and Baselines, ECCV 2024.

---

### Author Response · Authors · 2024-09-18
**General Comment**

We thank all the reviewers for their constructive comments and for appreciating our contributions. All reviewers commended our extensive experiments on 19 detectors and 5 different detection tasks. Specifically, Reviewer 1nWj mentioned “the work has extensive results with different datasets and models and usecases”; yiuu praised our paper for its ease of reading; and iRbf commended both robust experimental design and the presentation.

We have submitted a revision of our paper, and our revisions are written in blue color in the submitted version to make it easy to see the difference. In summary, considering the reviewers' comments;
- We included more detailed discussion on LaECE in Section 3.1.
- We made Refining NMS in Section 3.2 more clear and self-contained. As a result we removed Appendix D, which was providing the details on Refining NMS.
- We included Table 4 to show that calibrators also improve recent transformer-based object detectors.
- We included Table 5 to show that calibration and Refining NMS are useful for combining recent transformer-based object detectors. Section 4.1 presents the discussion on Table 4 and 5.
- We included Table 12 to show that MoCaE performs well also on recent instance segmentation models on COCO instance segmentation dataset, which was not previously used in the first submission. Section 4.2 includes a discussion on Table 12.
- We included the reliability diagrams of 13 different detectors in Figures A.10-A.14 in the Appendix to show the confidence score distribution of the detectors before and after calibration.

In the following, we provide detailed explanations for these revisions while addressing the individual concerns of each reviewer, alleviating any misconceptions there maybe.

---

### Author Response · Authors · 2024-10-08
**Gentle nudge**

Dear Reviewers,

We would like to thank you once again for your reviews and valuable time. We appreciate the comments you provided.

We have done our best to address all your concerns, including additional experiments and clarifications. We also reflected the necessary changes based on your reviews to the draft and submitted the revised versions of the paper during discussion.

In this regard, we request you to kindly have a look at our responses and please let us know if there are any further queries from your end. We believe that we have provided satisfactory responses and would be happy to address further questions if any.

Thank you.

---

### Decision · Action_Editor_1khr · 2024-10-11

**Recommendation:** Accept with minor revision

**Comment:**

Initially, the reviewers appreciated the clear motivation (yiuu), the simplicity of the approach (1nWj), the experimental results (iRbf, yiuu, 1nWj) and the presentation (iRbf, yiuu, 1nWj). However, they also raised concerns about the limited novelty (iRbf, yiuu) especially w.r.t. other works considering calibration for object detection (Kuzucu et al. 2024), the variety of experts tested (iRbf, yiuu, 1nWj), additional baselines (yiuu, 1nWj), clarifying parts of the discussion (1nWj), and potential data leakage on one benchmark (yiuu).

The authors' response clarified most of the concerns, especially those regarding including more experts/additional baselines, the discussion parts, and the potential leakage. 2/3 reviewers recommended acceptance, but 2/3 still raised concerns about the limited technical contribution as calibration has been already discussed in this field (Kuzucu et al. 2024) even if not in the context of the mixture of experts.

The AE went over reviews, answers, and updated manuscript and found the article to meet the criteria for acceptance to TMLR as 1) it clearly supports the statements and 2) despite the limited technical contribution, it is of interest to anyone who wants to develop mixture-of-experts models for object detection. Therefore, the AE recommends the acceptance of the work.

As a (very) minor change in the final version, it would be helpful to point out within the introduction how this work contributes to the discussion and studies of the calibration of object detectors. This discussion is mostly included at the end of Appendix A, but readers may still find (even only part of) it interesting to better contextualize the contribution of the work.

**Audience:**

As the mixture of experts is a powerful tool to improve performance on downstream tasks, showing that their calibration is important for object detection is definitely of interest to anyone who is developing models for this specific task. The fact that it is not explicitly shown how these principles may generalize to other tasks (i.e., beyond object detection and segmentation) limits the applicability of the findings (and the audience). Nevertheless, the core message remains interesting for practitioners and researchers in object detection and shows how model calibration should not be overseen.

**Claims And Evidence:**

The work presents MoCaE, a method that exploits a mixture of calibrated experts for object detection. Specifically, the article analyses how different object detectors can be individually calibrated and how their confidence can be used to improve object detection results. The paper's main claim is that calibrating the expert is the key element for reliably combining multiple detectors. While calibrating detectors is already something present in previous works (i.e., Kuzucu et al. 2024), this paper explicitly studied its application to the mixture of experts for object detection.

This main claim is well supported throughout the paper with multiple analyses and various benchmarks (in and out of domain), consistently showing that calibration improves w.r.t. the same mixture of experts but not calibrated.